# Epistemic Uncertainty Quantification
# for Pre-trained VLMs via Riemannian Flow Matching

Li Ju [1 2 *]   Mayank Nautiyal [1 2 *]   Andreas Hellander [1]   Ekta Vats [1]   Prashant Singh [1 2]

## Abstract

Vision-Language Models (VLMs) are typically deterministic in nature and lack intrinsic mechanisms to quantify epistemic uncertainty, which reflects the model's lack of knowledge or ignorance of its own representations. We theoretically motivate negative log-density of an embedding as a proxy for the epistemic uncertainty, where low-density regions signify model ignorance. The proposed method REPVLM computes the probability density on the hyperspherical manifold of the VLM embeddings using Riemannian Flow Matching. We empirically demonstrate that REPVLM achieves near-perfect correlation between uncertainty and prediction error, significantly outperforming existing baselines. Beyond classification, we also demonstrate that the model also provides a scalable metric for out-of-distribution detection and automated data curation.

## 1. Introduction

Vision-Language Models (VLMs) like CLIP (Radford et al., 2021), BLIP (Li et al., 2022), and SigLIP (Zhai et al., 2023) have significantly advanced cross-modal representation learning by aligning visual and textual data through large-scale contrastive pre-training. Their ability to map diverse modalities into a shared embedding space has enabled powerful zero-shot generalization across a variety of downstream tasks (Zhou et al., 2023; Wortsman et al., 2022). However, a fundamental limitation of these deterministic pre-trained models is that they map inputs to single, fixed points in a high-dimensional embedding space. This approach provides no intrinsic mechanism to represent the model's internal uncertainty of its own representations

(Chun et al., 2021; Chun, 2024; Chun et al., 2025).

While recent uncertainty quantification (UQ) frameworks for VLMs have emerged, they predominantly target aleatoric uncertainty (data ambiguity) through probabilistic embeddings (Chun et al., 2025; Baumann et al., 2024; Ju et al., 2025; Venkataramanan et al., 2025). Existing strategies for epistemic uncertainty (model ignorance), such as deep ensembles (Lakshminarayanan et al., 2017) or Monte Carlo dropout (Gal & Ghahramani, 2016), remain either computationally prohibitive for large-scale pre-trained models or yield suboptimal, batch-dependent estimates. Thus, there is a pressing need for a scalable, intrinsic metric that directly quantifies a model's confidence of its own representations.

We propose a principled proxy to quantify the epistemic uncertainty of pre-trained VLMs by estimating the probability density $p(z)$ of the embeddings directly on their native manifold. We posit that a low density indicates a region sparsely populated by the training data, signaling high epistemic uncertainty. Thus the negative log-likelihood, $-\log p(z)$ can serve as a robust and intrinsic uncertainty score. Further, we propose REPVLM (illustrated in Figure 1), a unified Conditional Riemannian Flow Matching (CRFM) framework that leverages the hyperspherical geometry $\mathbb{S}^{d-1}$ of $\ell_2$-normalized VLM embedding spaces to learn a modality-conditioned vector field, which can be used to compute the log-density of the embeddings.

Our contributions are summarized as follows:

- **Theoretical Motivation.** We establish a theoretical motivation demonstrating that the negative log-density $-\log p(z)$ can serve as a principled proxy for epistemic uncertainty of models based on training data coverage.
- **Manifold-Native Modeling.** We propose REPVLM, an epistemic UQ method that extends Riemannian Flow Matching to embedding spaces for scalable and intrinsic probability density computation for both visual and textual modalities.
- **Empirical Validation.** We demonstrate that REPVLM significantly outperforms baselines in selective classification, achieving near-perfect correlation with model error, and provides a reliable metric for out-of-distribution (OOD) detection and automated data curation.

---

[*]Equal contribution  [1]Department of Information Technology, Uppsala University, Uppsala, Sweden [2]Science for Life Laboratory, Uppsala University, Uppsala, Sweden. Correspondence to: Li Ju <li.ju@it.uu.se>.

*Proceedings of the $43^{rd}$ International Conference on Machine Learning*, Seoul, South Korea. PMLR 306, 2026. Copyright 2026 by the author(s).

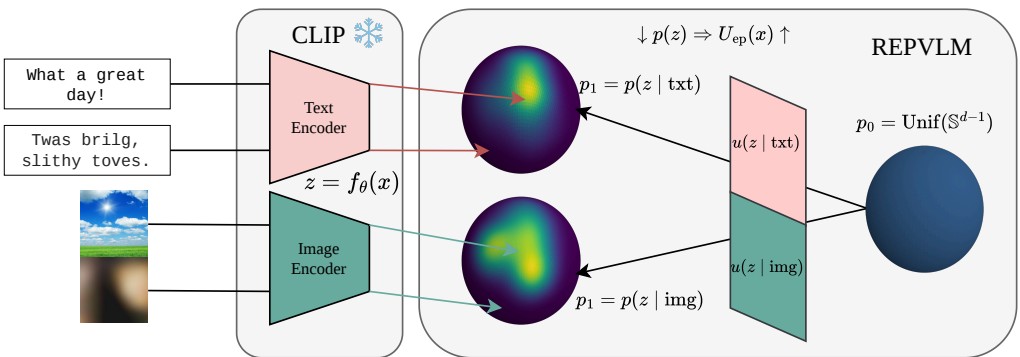

*Figure 1.* **Overview of REPVLM.** The framework estimates the probability density $p(z)$ of $\ell_2$-normalized pre-trained VLM embeddings on the hypersphere $\mathbb{S}^{d-1}$. A unified model learns a vector field $v_t$ that transports a simple uniform base distribution $P_0 = \text{Unif}(\mathbb{S}^{d-1})$ to the empirical modality-specific distributions $P_1$ (Image and Text). As illustrated, standard inputs map to high-density regions (yellow), while ambiguous or out-of-distribution inputs such as the distorted image or nonsensical text reside in low-density regions (purple). The negative log-likelihood $-\log p(z|c)$ thus serves as a principled proxy for epistemic uncertainty $U_{\text{ep}}(z)$, reflecting the model's confidence.

**Outline** Section 2 reviews related work in UQ for VLMs and Flow Matching (FM). Section 3 motivates the link between latent density and epistemic uncertainty. Section 4 introduces REPVLM, detailing the geometry-aware problem formulation and training objective. Section 5 presents empirical results and Section 6 concludes with a discussion of limitations.

## 2. Related Work

**VLMs** Modern VLMs learn a shared semantic embedding space by aligning visual and textual representations. Foundational models such as CLIP (Radford et al., 2021), BLIP (Li et al., 2022), and SigLIP (Zhai et al., 2023) achieve this through large-scale contrastive pre-training, optimizing an objective based on the similarity between paired image and text embeddings. While this paradigm has demonstrated powerful zero-shot generalization, a fundamental structural limitation is that these models typically map inputs to single, fixed points in the embedding space. This approach provides no intrinsic mechanism to represent the model's internal confidence or uncertainty regarding its own representations. Consequently, it is difficult to assess the reliability of VLM outputs, particularly when the model is presented with ambiguous or out-of-distribution samples that it may not truly "understand" (Zhang et al., 2024).

**UQ for VLMs** Table 1 compares existing methods for UQ in VLMs. Current methods predominantly address aleatoric uncertainty (data ambiguity). This is typically achieved by learning probabilistic embeddings, either by training models from scratch (Chun et al., 2021; Chun, 2024; Chun et al., 2025) or by adapting pre-trained ones (Upadhyay et al., 2023; Baumann et al., 2024; Ju et al., 2025; Venkataramanan et al., 2025). While effective for modeling data ambiguity,

these approaches lack a principled measure of epistemic uncertainty for embeddings (model confidence).

Alternatively, Gomez (2025) quantifies uncertainty in cross-modal retrieval by measuring the minimal perturbation required to alter result. While providing a practical measure for retrieval, it introduces challenges for general-purpose UQ. Specifically, because the estimate is dependent on the particular batch of target used as candidates, it is not an intrinsic property of the query itself. Ideally, an uncertainty measure should depend exclusively on the input and the model's state. Furthermore, the theoretical gap between this perturbation-based metric to formal definitions of uncertainty remains for further clarification. To the best of our knowledge, ProbVLM (Upadhyay et al., 2023) is one of the few existing approaches for quantifying the epistemic uncertainty of pre-trained VLMs. Alternatively, by enabling Monte Carlo dropout of the pre-trained models during the inference stage with multiple passes, the epistemic uncertainty can also be estimated through the variance of the resulting embeddings (Gal & Ghahramani, 2016).

**Density and geometric structure of contrastive embedding spaces** A parallel line of work analyses the latent space of contrastive VLMs to extract signals beyond their discriminative role: Betser et al. (2025) use a Gaussian likelihood on whitened CLIP embeddings as a likelihood surrogate; Levi & Gilboa (2025) characterise the global geometry as a double-ellipsoid and derive a conformity score; Draganov et al. (2025) show that the embedding norm carries information about training-set density; and Liang et al. (2022) document the modality gap that any modality-conditional density model must accommodate. None of these measures were designed for UQ, but under our framing each can be repurposed as a confidence score – as a density surrogate (W-CLIP), a geometry-derived conformity

*Table 1.* Comparison of methods for uncertainty quantification for vision language models.

| Method | Post-hoc | $U_{al}(z)$ | $U_{ep}(z)$ | Based on |
|---|---|---|---|---|
| Full Dropout | ✓ | ✗ | ✓ | MC Dropout |
| PFE | ✗ | ✓ | ✗ | Prob. Embed. |
| PCME+ | ✗ | ✓ | ✗ | Prob. Embed. |
| ProLIP | ✗ | ✓ | ✗ | Prob. Embed. |
| BayesVLM | ✓ | ✓ | ✗ | Prob. Embed. |
| AsymVLM | ✓ | ✓ | ✗ | Prob. Embed. |
| Adv. Pert | ✓ | ? | ? | Perturbation |
| GroVE | ✓ | ✓ | ✗ | Prob. Embed. |
| ProbVLM | ✓ | ✓ | ✓ | Prob. Embed. +MC Dropout |
| **Ours** | ✓ | ✗ | ✓ | Prob. Density |

measure (Conformity), or a scalar density-related summary (Embedding Norm). We include them as additional baselines, deferring the empirical comparison to Section 5.

**Normalizing Flows** Chen et al. (2018) offer exact density estimation by transforming simple base distributions into complex distributions through invertible mappings. However, the requirement for a tractable Jacobian determinant imposes significant computational cost for training (Grathwohl et al., 2018). Flow Matching (Lipman et al.) has emerged as a more efficient, simulation-free method for training Continuous Normalizing Flows. Instead of learning an invertible map, FM learns a vector field that defines a probability flow between distributions, improving training stability and efficiency. This framework retains the ability to compute exact log-likelihoods by integrating the divergence of the learned vector field along an ordinary differential equation trajectory. Crucially, its recent extension Riemannian Flow Matching (Chen & Lipman, 2024) generalizes this approach to non-Euclidean manifolds, allowing for probability modeling directly on the hypersphere $\mathbb{S}^{d-1}$ where $\ell_2$-normalized VLM embeddings reside.

## 3. Rethinking Uncertainty Quantification

While the distinction between aleatoric and epistemic uncertainty is well-established (Hüllermeier & Waegeman, 2021), existing VLM frameworks struggle to decouple them effectively. Current state-of-the-art methods predominantly focus on aleatoric uncertainty by learning probabilistic embeddings, i.e., mapping inputs to distributions to capture data ambiguity. However, these methods often fail to provide a calibrated measure of the model's own "ignorance" (Upadhyay et al., 2023; Baumann et al., 2024).

Standard epistemic estimators like Bayesian Neural Networks (Blundell et al., 2015) or deep ensembles (Lakshminarayanan et al., 2017) remain computationally prohibitive

at modern VLM-scale. Meanwhile, common approximations such as Monte Carlo Dropout (Upadhyay et al., 2023; Srivastava et al., 2014) or heuristic perturbations (Gomez, 2025) often yield suboptimal or batch-dependent estimates.

> We propose an alternative **hypothesis:** Epistemic uncertainty is reflected in the local density of an embedding on the embedding manifold. Estimating the probability density of the VLM's embedding allows us to move away from expensive parameter-space sampling, toward a principled, scalable proxy for epistemic uncertainty.

In this setting, an embedding located in a region sparsely populated by training data (low $p(z)$) signals high epistemic uncertainty. Thus, the negative log-likelihood, $-\log p(z)$, serves as a principled, scalable proxy for the model's epistemic uncertainty.

### 3.1. Theoretical Motivation

We motivate using $-\log p(z)$ as a proxy for $U_{ep}(x)$ via a three-link chain. To avoid overclaiming, we tag each link by its character: **(A1)** is a formal derivation under a standard Bayesian DL assumption (Immer et al., 2021); **(A2)** is an expectation-level optimality condition; **(A3)** is an empirical assumption specific to contrastive loss, motivated by alignment-and-uniformity dynamics (Wang & Isola, 2020).

**Definition 3.1** (Epistemic Uncertainty). The epistemic uncertainty of a Bayesian encoder $f(x; \theta)$ for a given input $x$ is defined as the trace of the covariance matrix of its embedding $z = f(x; \theta)$ over the posterior distribution of the parameters $p(\theta|D)$:

$$U_{ep}(x) = \text{Tr}(\text{Cov}_{p(\theta|D)}(z))$$
$$= \text{Tr}\left(\mathbb{E}_{p(\theta|D)}\left[(f(x;\theta) - \bar{z})(f(x;\theta) - \bar{z})^\top\right]\right),$$

where $\bar{z} = \mathbb{E}_{p(\theta|D)}[f(x;\theta)]$ represents the mean embedding over the parameter posterior $p(\theta|D)$.

**Assumption 3.2** (Local Linearity of the Encoder). We then assume that the encoder function $f(x; \theta)$ is locally linear with respect to its parameters $\theta$ within the high-density region of the posterior surrounding the maximum a posteriori estimate $\theta^*$. This can be expressed via a first-order Taylor expansion around $\theta^*$:

$$f(x;\theta) \approx f(x;\theta^*) + J_{\theta^*}(x)(\theta - \theta^*),$$

where $J_{\theta^*}(x) = \frac{\partial f(x;\theta)}{\partial \theta}\big|_{\theta=\theta^*}$ is the Parameter-Jacobian of the encoder at $\theta^*$.

**Covariance to Jacobian Norm (A1)** Given that the expectation of the linearized function is $\mathbb{E}_{p(\theta|D)}[f(x;\theta)] \approx f(x;\theta^*)$, under Assumption 3.2 the covariance of the embedding $z$ is:

$$\text{Cov}(z) \approx \mathbb{E}_{p(\theta|D)}\left[(J_{\theta^*}(x)(\theta - \theta^*))(J_{\theta^*}(x)(\theta - \theta^*))^\top\right]$$
$$= J_{\theta^*}(x)\Sigma_\theta J_{\theta^*}(x)^\top,$$

where $\Sigma_\theta = \text{Cov}(\theta)$ is the parameter covariance. The epistemic uncertainty, given by the trace of this matrix, is thus proportional to a quadratic form of the Parameter-Jacobian: $U_{\text{ep}}(x) \approx \text{Tr}(J_{\theta*}(x)\Sigma_\theta J_{\theta*}(x)^\top)$. This establishes our first link: high epistemic uncertainty corresponds to a large Parameter-Jacobian norm, indicating that the model's output is sensitive to perturbations in its learned parameters. The proportionality is exact only in the linearized regime; we use it as a first-order approximation throughout.

**Jacobian Norm to Data Density (A2)** The second link connects $\|J_\theta(x)\|_F$ to the training data density $p(x)$. For an encoder trained by minimizing the risk $R = \mathbb{E}_{x \sim P_x}[\ell(f(x; \theta), x)]$, the first-order optimality condition $\nabla_\theta R(\theta^*) = 0$ implies:

$$\int (\nabla_z \ell)^\top J_\theta(x) \, p(x) \, dx = 0.$$

This is an *expectation-level* condition and does not by itself imply pointwise suppression of $\|J_\theta(x)\|_F$ in high-density regions. In practice, the smoothness of the encoder produces a clear point-wise tendency: in-distribution samples concentrate where $\|J_\theta(x)\|_F$ is small, while for OOD inputs $x_{\text{out}}$ with $p(x_{\text{out}}) \approx 0$ the Jacobian norm remains largely unconstrained.

**Data Density to Embedding Density (A3)** The final step translates $p(x)$ to the computable embedding density $p(z)$. Being non-invertible and reducing dimension, the encoder is not a strict isometry, and establishing a formal density-preservation result for contrastive encoders is a recognized open problem in representation learning (Zimmermann et al., 2021; Betser et al., 2026; Cai et al., 2026; Huh et al., 2024). We therefore treat $p(z) \propto p(x)$ as an empirical assumption, motivated by alignment-and-uniformity dynamics (Wang & Isola, 2020; Saxe et al., 2014): semantically similar inputs are mapped to tight clusters on $\mathbb{S}^{d-1}$, while rare or OOD inputs are pushed to sparse regions. Independent empirical support comes from whitened CLIP embeddings serving as a likelihood surrogate (Betser et al., 2025).

**Chain Summary** Composing the three links:

$$\uparrow p(z) \xrightarrow{\text{(A3)}} \uparrow p(x) \xrightarrow{\text{(A2)}} \downarrow \|J_\theta(x)\|_F \xrightarrow{\text{(A1)}} \downarrow U_{\text{ep}}(x).$$

Under (A1)–(A3), $-\log p(z)$ is a *monotone proxy* for $U_{\text{ep}}(x)$. This allows us to quantify model confidence without the intractable computation of the parameter posterior for large-scale VLMs.

### 3.2. Why Flow Matching

To estimate the density over the VLM's embedding space, the chosen method must satisfy three critical criteria: scalability to high dimensions, efficiency in evaluation, and geometric compatibility with the manifold. Flow Matching (Lipman et al.) and its Riemannian extension (RFM) (Chen & Lipman, 2024) are uniquely suited for this task as:

- Unlike standard Normalizing Flows, FM avoids costly Jacobian determinant computations. It enables efficient training for high-dimensional VLM embeddings by learning a vector field.
- RFM models densities directly on the hypersphere $\mathbb{S}^{d-1}$. Utilizing geodesic interpolation rather than Euclidean paths allows the model to respect the intrinsic geometry of the semantic space.
- RFM computes the exact log-density by integrating the divergence of the vector field. Leveraging Hutchinson's trace estimator, the integration can be approximated by a few steps, ensuring minimal inference overhead.

## 4. Method

We propose REPVLM, a framework to quantify epistemic uncertainty by estimating the probability density of pre-trained VLM embeddings. Building upon CRFM, we learn the embedding distributions of both modalities within a unified neural network.

### 4.1. Problem Formulation and Geometry

Let $\mathcal{Z} \subseteq \mathbb{S}^{d-1}$ denote the $d$-dimensional shared embedding space of a pre-trained VLM. For input $x$, the encoder $f_\theta$ produces a $\ell_2$ normalized embedding $z = f_\theta(x)$.

Following the theoretical grounding in Section 3, we define the epistemic uncertainty $U_{\text{ep}}(x)$ as the negative log-likelihood of the embedding conditioned on its modality:

$$U_{\text{ep}}(x) = -\log p(z \mid c),$$

where $c \in \{0, 1\}$ serves as a discrete conditioning variable (e.g., $c = 0$ for images, $c = 1$ for text). To maintain geometric consistency, all updates must respect the tangent space $T_z \mathbb{S}^{d-1} = \{v \in \mathbb{R}^d : \langle v, z \rangle = 0\}$ via the projection operator $\Pi_{T_z}(v) = (I - zz^\top)v$.

### 4.2. Conditional Riemannian Flow Matching

We define a conditional probability path $p_t(z \mid c)$ for $t \in [0, 1]$ that transforms a uniform base distribution into the target empirical distribution for modality $c$.

#### 4.2.1. UNIFIED CONDITIONAL RIEMANNIAN FLOW

We parameterize a single time-dependent vector field $v_t(z, c; \phi)$ using a neural network with the embedding $z$, time $t$ and modality indicator $c$ as inputs. The flow is defined by the conditional Ordinary Differential Equation (ODE):

$$\frac{dz_t}{dt} = v_t(z_t, c; \phi), \quad z_0 \sim p_0.$$

The boundary conditions are defined as follows:

- **Base Distribution** $p_0$**:** A modality-agnostic uniform distribution over the hypersphere, $p_0 = \text{Unif}(\mathbb{S}^{d-1})$.
- **Target Distribution** $p_1 = p(z \mid c)$**:** The empirical distribution of VLM embeddings for modality $c$, estimated from a *proxy dataset* of image-caption pairs (i.e., $p(z \mid c = 0) = p_{\text{image}}(z)$ and $p(z \mid c = 1) = p_{\text{text}}(z)$). Crucially, this proxy dataset is distinct from downstream task, enabling task-agnostic uncertainty estimation.

### 4.2.2. GEODESIC CONDITIONAL PATH

Unlike Euclidean flow matching, we construct paths along geodesics. Given a base sample $z_0 \sim p_0$ and a target sample $z_1 \sim p(z \mid c)$, the sample-conditioned path follows:

$$z_t = \frac{\sin((1-t)\theta)}{\sin\theta} z_0 + \frac{\sin(t\theta)}{\sin\theta} z_1,$$

where $\theta = \arccos(\langle z_0, z_1 \rangle)$ is the geodesic distance between the two points. The target vector field $u_t(z_t \mid z_1)$ is the tangent to the geodesic at time $t$ (in closed-form):

$$u_t(z_t \mid z_1) = \frac{\theta}{\sin\theta} \big( \cos(t\theta)z_1 - \cos((1-t)\theta)z_0 \big).$$

### 4.2.3. TRAINING OBJECTIVE

The unified network is trained using the RFM objective, regressing the vector field onto the target $u_t(z_t \mid z_1, c)$:

$$\mathcal{L}(\phi) = \mathbb{E}_{p(c),p(t),p(z_1),p(z_t)}\big[\|v_t(z_t, c; \phi) - u_t(z_t \mid z_1, c)\|^2\big],$$

where $c \sim \text{Bernoulli}(0.5), t \sim \text{Unif}([0,1]), z_1 \mid c \sim p(z \mid c)$ and $z_t \sim p_t(z_t \mid c)$. By conditioning on $c$, the network $\phi$ learns to disentangle the text and image embedding geometries while sharing parameters where appropriate, since both modalities are designed to embed the same semantic space. The algorithm is summarized as Algorithm 1 in Appendix.

## 4.3. Inference and Uncertainty Quantification

To quantify the epistemic uncertainty for an embedding $z$ of modality $c$, we estimate its exact log-density by integrating the divergence of the learned conditional vector field.

### 4.3.1. LOG-LIKELIHOOD VIA CONTINUITY EQUATION

Using the instantaneous change of variables formula on the manifold, the log-density at the target distribution is:

$$\log p_1(z \mid c) = \log p_0(\psi_0(z)) - \int_0^1 \text{div}_{\mathbb{S}^{d-1}}(v_t(z_t, c; \phi)) \, dt,$$

where $\psi_t$ denotes the flow map induced by integrating the ODE backward from $t = 1$ to $t = 0$. Since the base distribution is uniform, its log-density is constant: $\log p_0(z) = -\log \text{Vol}(\mathbb{S}^{d-1})$, which simplifies the likelihood computation to integrating only the divergence term.

**Manifold-Aware ODE Integration** We solve the reverse ODE using a Riemannian-adapted Euler's method. At each integration step, intermediate states are projected back onto $\mathbb{S}^{d-1}$ via normalization, and velocity evaluations are projected onto the tangent space $T_z\mathbb{S}^{d-1}$. This ensures numerical integrity and prevents drifting away from the manifold over many integration steps.

**Riemannian Divergence Estimation** Computing the exact divergence $\text{div}_{\mathbb{S}^{d-1}}(v_t)$ is computationally expensive for high-dimensional VLM embeddings as it requires $\mathcal{O}(d)$ backward passes. Instead, we employ a manifold-constrained Hutchinson trace estimator (Hutchinson, 1989):

$$\text{div}_{\mathbb{S}^{d-1}}(v_t) \approx \langle \tilde{\epsilon}, \nabla_z(v_t \cdot \tilde{\epsilon}) \rangle,$$
$$\text{where } \tilde{\epsilon} = \Pi_{T_z}(\epsilon), \ \epsilon \sim \mathcal{N}(0, I).$$

$\Pi_{T_z}(v) = v - \langle v, z \rangle z$ projects the random probe vector $\epsilon$ onto the tangent space at $z$. This allows for an unbiased estimate of the divergence with significant reduction in computational cost.

**Epistemic Uncertainty Score** The resulting log-density yields the epistemic uncertainty score:

$$U_{\text{ep}}(x) = -\log p_1(z \mid c).$$

High uncertainty (low density) identifies samples that reside in sparse regions of the manifold, such as outliers, noisy inputs, and nonsensical content.

A more comprehensive derivation including the construction of the probability path, the training objective, and the calculation of the likelihood, is deferred to Appendix A.

## 5. Empirical results

We evaluate REPVLM's ability to quantify epistemic uncertainty through selective classification on benchmark datasets first. Then, we provide an ablation study for REPVLM. Further, we show two applications of our uncertainty estimates including out-of-distribution detection and data curation.

### 5.1. Datasets, baselines, and metrics

We train REPVLM on three proxy datasets: Conceptual Captions (Sharma et al., 2018), DataComp-1B (Gadre et al., 2023), and LAION-2B (Schuhmann et al., 2022). To ensure computational efficiency, we randomly sample 1M pairs from each dataset. For evaluation, we assess zero-shot classification across six benchmarks: ImageNet-1K (Deng et al., 2009), Food101 (Bossard et al., 2014), Cifar100 (Krizhevsky et al., 2009), ObjectNet (Barbu et al., 2019), ImageNet-R (Kornblith et al., 2019), and ImageNet-Sketch (Wang et al., 2019). All experiments reported in the main manuscript use OpenCLIP with a ViT-B/32 backbone

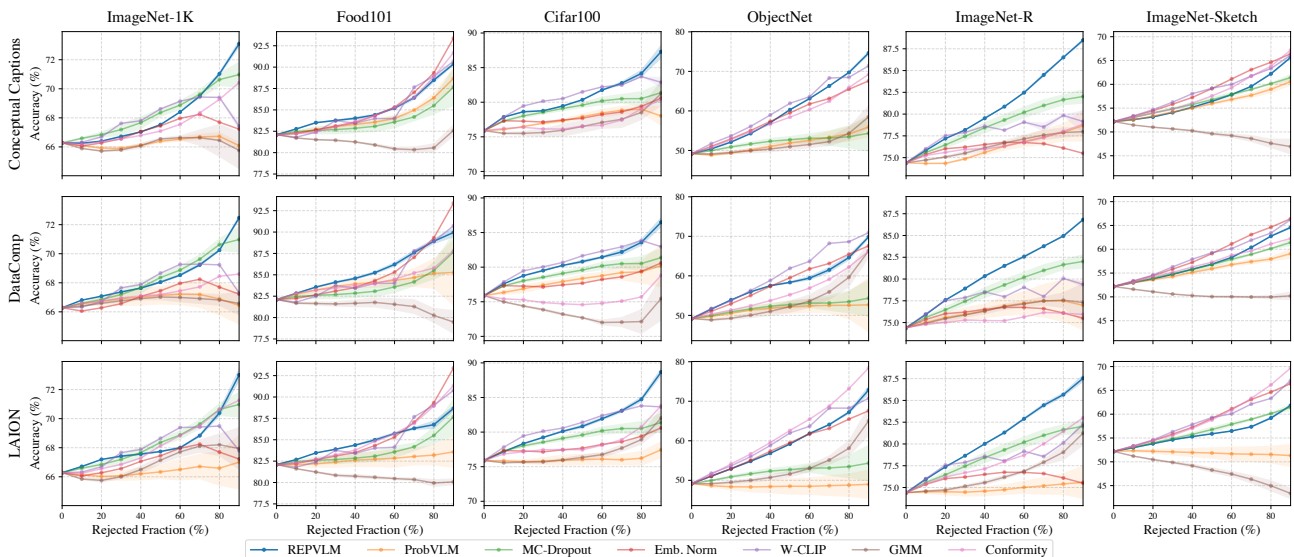

Figure 2. **Accuracy-Rejection Curves for Zero-Shot Classification.** We evaluate the utility of epistemic uncertainty estimates for selective classification across three proxy datasets (rows) and six downstream benchmarks (columns). The $x$-axis represents the fraction of samples rejected based on high uncertainty, and the $y$-axis shows the Top-1 Accuracy of the remaining retained data. A sharp upward slope indicates that the model identifies its own errors via the uncertainty measure.

(Cherti et al., 2023). Results on more backbone encoders are provided in Appendix B.6.

We compare REPVLM against two existing methods for epistemic uncertainty quantification for frozen VLMs, Prob-VLM (Upadhyay et al., 2023) and Monte Carlo Dropout (MCDO), together with four additional baselines repurposed as confidence scores from the embedding-density and geometry literature discussed in Section 2: Embedding Norm (Draganov et al., 2025), the pre-normalisation $\ell_2$ norm of the embedding; W-CLIP (Betser et al., 2025), the Gaussian negative log-likelihood of the whitened embedding; Conformity (Levi & Gilboa, 2025), the cosine similarity between the embedding and its modality-specific mean, and a Gaussian Mixture Model (GMM) fitted to the proxy embeddings as a Euclidean parametric density baseline. Aleatoric-only methods listed in Table 1 are excluded as they do not provide a principled measure of model confidence.

We utilize selective classification (Geifman & El-Yaniv, 2017) to measure the effectiveness of the uncertainty scores, reporting the following metrics:

- **Accuracy-Rejection Curves**: These plot Top-1 accuracy as the most uncertain samples (0–90%) are progressively rejected; an upward slope indicates the model successfully identifies its own errors via uncertainty measures.
- **Acc@90% Rejection**: Top-1 accuracy on the 10% most confident predictions.
- **Spearman's** $S$: The correlation between the rejection fraction and accuracy. An $S = 1.0$ indicates a perfect monotonic relationship where increasing the rejection

threshold consistently improves accuracy

All experiments were repeated five times with different random seeds, and we report the mean with standard deviations. The code for reproducing our results is available at https://github.com/li-ju666/repvlm.

### 5.2. Main results

The numerical results of the epistemic uncertainty evaluation are reported in Table 2, with the corresponding accuracy-rejection curves shown in Figure 2.

**Reliability on Standard Benchmarks** On the standard benchmarks (ImageNet-1K, Food101, Cifar100), REPVLM attains near-perfect Spearman correlation ($S \geq 0.988$) in every cell and the best Acc@90 on ImageNet-1K and Cifar100 across all three proxies; the only exception is Food101, where Embedding Norm leads on Acc@90 (0.934) while REPVLM ties it on Spearman ($S \geq 0.998$). Conformity is the closest density-aware competitor, also reaching $S = 1.000$ on most cells but trailing on Acc@90 (e.g. 0.686–0.713 vs. 0.725–0.731 on ImageNet-1K) and collapsing on Cifar100/DataComp ($S = 0.042$); ProbVLM and GMM are uneven, with ProbVLM's Spearman degrading on DataComp/LAION (down to $S = 0.290$) and GMM's flipping negative on Food101 across all proxies (down to $S = -0.908$). REPVLM's margin is widest on the cleaner proxies and narrows on LAION – a structural feature of any density-based UQ method, since the estimated density inherits the cleanliness of the proxy. We discuss this proxy-dependence in Section 6.

*Table 2.* Evaluation of model performance and uncertainty on benchmarks across different training configurations (Conceptual Captions, DataComp, and LAION). For each method, we report Accuracy at 90% Rejection ↑ and Spearman's rank correlation $S$ ↑ between uncertainty and prediction error. **Bold** font denotes the best results. Note: MCDO and Embed. Norm do not depend on the proxy dataset; columns repeat for ease of comparison.

| | | CONCEPTUAL CAPTIONS | | DATACOMP | | LAION | |
|---|---|---|---|---|---|---|---|
| EVAL. ON | METHOD | Acc. @ 90% Rej. ↑ | $S$ ↑ | Acc. @ 90% Rej. ↑ | $S$ ↑ | Acc. @ 90% Rej. ↑ | $S$ ↑ |
| ImageNet-1K | MC-Dropout | $0.710 \pm 0.009$ | $\mathbf{0.995 \pm 0.006}$ | $0.710 \pm 0.009$ | $0.995 \pm 0.006$ | $0.710 \pm 0.009$ | $\mathbf{0.995 \pm 0.006}$ |
| | ProbVLM | $0.661 \pm 0.003$ | $0.491 \pm 0.178$ | $0.664 \pm 0.013$ | $0.290 \pm 0.585$ | $0.670 \pm 0.019$ | $0.421 \pm 0.738$ |
| | Emb. Norm | $0.672 \pm 0.000$ | $0.806 \pm 0.000$ | $0.672 \pm 0.000$ | $0.806 \pm 0.000$ | $0.672 \pm 0.000$ | $0.806 \pm 0.000$ |
| | W-CLIP | $0.674 \pm 0.003$ | $0.762 \pm 0.058$ | $0.673 \pm 0.001$ | $0.707 \pm 0.005$ | $0.678 \pm 0.001$ | $0.796 \pm 0.052$ |
| | GMM | $0.658 \pm 0.013$ | $0.355 \pm 0.465$ | $0.666 \pm 0.008$ | $0.350 \pm 0.535$ | $0.679 \pm 0.015$ | $0.789 \pm 0.193$ |
| | Conformity | $0.704 \pm 0.000$ | $\underline{0.988 \pm 0.000}$ | $0.686 \pm 0.000$ | $\mathbf{1.000 \pm 0.000}$ | $\underline{0.713 \pm 0.000}$ | $0.988 \pm 0.000$ |
| | REPVLM | $\mathbf{0.733 \pm 0.003}$ | $\underline{0.988 \pm 0.000}$ | $\mathbf{0.725 \pm 0.002}$ | $1.000 \pm 0.000$ | $\underline{0.730 \pm 0.005}$ | $0.995 \pm 0.006$ |
| Food101 | MC-Dropout | $0.877 \pm 0.022$ | $0.891 \pm 0.177$ | $0.877 \pm 0.022$ | $0.891 \pm 0.177$ | $0.877 \pm 0.022$ | $0.891 \pm 0.177$ |
| | ProbVLM | $0.887 \pm 0.013$ | $0.998 \pm 0.005$ | $0.853 \pm 0.039$ | $0.641 \pm 0.371$ | $0.836 \pm 0.019$ | $0.467 \pm 0.543$ |
| | Emb. Norm | $\mathbf{0.934 \pm 0.000}$ | $\mathbf{1.000 \pm 0.000}$ | $\mathbf{0.934 \pm 0.000}$ | $\mathbf{1.000 \pm 0.000}$ | $\mathbf{0.934 \pm 0.000}$ | $\mathbf{1.000 \pm 0.000}$ |
| | W-CLIP | $0.907 \pm 0.000$ | $0.971 \pm 0.006$ | $0.907 \pm 0.000$ | $0.976 \pm 0.000$ | $0.908 \pm 0.000$ | $0.976 \pm 0.000$ |
| | GMM | $0.826 \pm 0.007$ | $-0.479 \pm 0.109$ | $0.795 \pm 0.014$ | $-0.498 \pm 0.351$ | $0.801 \pm 0.004$ | $-0.908 \pm 0.154$ |
| | Conformity | $\underline{0.917 \pm 0.000}$ | $\mathbf{1.000 \pm 0.000}$ | $0.878 \pm 0.000$ | $\mathbf{1.000 \pm 0.000}$ | $\underline{0.913 \pm 0.000}$ | $\mathbf{1.000 \pm 0.000}$ |
| | REPVLM | $0.904 \pm 0.002$ | $\mathbf{1.000 \pm 0.000}$ | $0.899 \pm 0.006$ | $\mathbf{1.000 \pm 0.000}$ | $0.887 \pm 0.003$ | $0.998 \pm 0.005$ |
| Cifar100 | MC-Dropout | $0.814 \pm 0.014$ | $\underline{0.968 \pm 0.042}$ | $0.814 \pm 0.014$ | $0.968 \pm 0.042$ | $0.814 \pm 0.014$ | $0.968 \pm 0.042$ |
| | ProbVLM | $0.780 \pm 0.014$ | $\underline{0.816 \pm 0.199}$ | $0.802 \pm 0.031$ | $0.818 \pm 0.240$ | $0.774 \pm 0.012$ | $0.321 \pm 0.391$ |
| | Emb. Norm | $0.806 \pm 0.000$ | $0.952 \pm 0.000$ | $0.806 \pm 0.000$ | $0.952 \pm 0.000$ | $0.806 \pm 0.000$ | $0.952 \pm 0.000$ |
| | W-CLIP | $\underline{0.829 \pm 0.008}$ | $0.968 \pm 0.045$ | $0.830 \pm 0.000$ | $\underline{0.978 \pm 0.012}$ | $0.836 \pm 0.001$ | $\underline{0.988 \pm 0.000}$ |
| | GMM | $0.812 \pm 0.018$ | $0.799 \pm 0.169$ | $0.755 \pm 0.032$ | $-0.595 \pm 0.320$ | $0.820 \pm 0.016$ | $0.881 \pm 0.082$ |
| | Conformity | $0.804 \pm 0.000$ | $0.891 \pm 0.000$ | $0.788 \pm 0.000$ | $0.042 \pm 0.000$ | $\underline{0.838 \pm 0.000}$ | $0.964 \pm 0.000$ |
| | REPVLM | $\mathbf{0.873 \pm 0.010}$ | $\mathbf{1.000 \pm 0.000}$ | $\mathbf{0.865 \pm 0.008}$ | $\mathbf{1.000 \pm 0.000}$ | $\mathbf{0.887 \pm 0.006}$ | $\mathbf{1.000 \pm 0.000}$ |
| ObjectNet | MC-Dropout | $0.543 \pm 0.047$ | $0.743 \pm 0.472$ | $0.543 \pm 0.047$ | $0.743 \pm 0.472$ | $0.543 \pm 0.047$ | $0.743 \pm 0.472$ |
| | ProbVLM | $0.559 \pm 0.026$ | $0.959 \pm 0.053$ | $0.527 \pm 0.070$ | $0.505 \pm 0.535$ | $0.490 \pm 0.037$ | $-0.052 \pm 0.809$ |
| | Emb. Norm | $0.676 \pm 0.000$ | $\mathbf{1.000 \pm 0.000}$ | $0.676 \pm 0.000$ | $\mathbf{1.000 \pm 0.000}$ | $0.676 \pm 0.000$ | $\mathbf{1.000 \pm 0.000}$ |
| | W-CLIP | $\underline{0.714 \pm 0.000}$ | $\mathbf{1.000 \pm 0.000}$ | $\mathbf{0.709 \pm 0.000}$ | $\mathbf{1.000 \pm 0.000}$ | $0.708 \pm 0.000$ | $0.990 \pm 0.005$ |
| | GMM | $0.585 \pm 0.030$ | $0.971 \pm 0.052$ | $0.663 \pm 0.020$ | $0.973 \pm 0.026$ | $0.650 \pm 0.038$ | $0.983 \pm 0.028$ |
| | Conformity | $0.693 \pm 0.000$ | $\mathbf{1.000 \pm 0.000}$ | $0.662 \pm 0.000$ | $\mathbf{1.000 \pm 0.000}$ | $\underline{0.784 \pm 0.000}$ | $\mathbf{1.000 \pm 0.000}$ |
| | REPVLM | $\mathbf{0.746 \pm 0.006}$ | $\mathbf{1.000 \pm 0.000}$ | $0.699 \pm 0.005$ | $\mathbf{1.000 \pm 0.000}$ | $\underline{0.728 \pm 0.011}$ | $\mathbf{1.000 \pm 0.000}$ |
| ImageNet-R | MC-Dropout | $0.820 \pm 0.008$ | $0.998 \pm 0.005$ | $0.820 \pm 0.008$ | $0.998 \pm 0.005$ | $0.820 \pm 0.008$ | $0.998 \pm 0.005$ |
| | ProbVLM | $\underline{0.787 \pm 0.018}$ | $0.930 \pm 0.055$ | $\underline{0.770 \pm 0.029}$ | $0.702 \pm 0.554$ | $0.756 \pm 0.019$ | $0.277 \pm 0.718$ |
| | Emb. Norm | $0.755 \pm 0.000$ | $0.430 \pm 0.000$ | $0.755 \pm 0.000$ | $0.430 \pm 0.000$ | $0.755 \pm 0.000$ | $0.430 \pm 0.000$ |
| | W-CLIP | $0.792 \pm 0.013$ | $0.842 \pm 0.146$ | $0.794 \pm 0.004$ | $0.927 \pm 0.000$ | $0.822 \pm 0.001$ | $0.961 \pm 0.018$ |
| | GMM | $0.780 \pm 0.005$ | $0.987 \pm 0.014$ | $0.773 \pm 0.023$ | $0.840 \pm 0.204$ | $0.812 \pm 0.014$ | $0.998 \pm 0.005$ |
| | Conformity | $0.788 \pm 0.000$ | $\mathbf{1.000 \pm 0.000}$ | $0.759 \pm 0.000$ | $0.903 \pm 0.000$ | $\underline{0.830 \pm 0.000}$ | $\mathbf{1.000 \pm 0.000}$ |
| | REPVLM | $\mathbf{0.885 \pm 0.002}$ | $\mathbf{1.000 \pm 0.000}$ | $\mathbf{0.868 \pm 0.001}$ | $\mathbf{1.000 \pm 0.000}$ | $\mathbf{0.875 \pm 0.004}$ | $\mathbf{1.000 \pm 0.000}$ |
| ImageNet-Sketch | MC-Dropout | $0.614 \pm 0.008$ | $\mathbf{1.000 \pm 0.000}$ | $0.614 \pm 0.008$ | $\mathbf{1.000 \pm 0.000}$ | $0.614 \pm 0.008$ | $\mathbf{1.000 \pm 0.000}$ |
| | ProbVLM | $0.606 \pm 0.009$ | $\mathbf{1.000 \pm 0.000}$ | $0.591 \pm 0.011$ | $0.993 \pm 0.015$ | $0.513 \pm 0.024$ | $0.035 \pm 0.871$ |
| | Emb. Norm | $\underline{0.664 \pm 0.000}$ | $\mathbf{1.000 \pm 0.000}$ | $\mathbf{0.664 \pm 0.000}$ | $\mathbf{1.000 \pm 0.000}$ | $0.664 \pm 0.000$ | $\mathbf{1.000 \pm 0.000}$ |
| | W-CLIP | $0.661 \pm 0.004$ | $\mathbf{1.000 \pm 0.000}$ | $0.662 \pm 0.001$ | $\mathbf{1.000 \pm 0.000}$ | $0.671 \pm 0.004$ | $\mathbf{1.000 \pm 0.000}$ |
| | GMM | $0.469 \pm 0.017$ | $-0.993 \pm 0.015$ | $0.502 \pm 0.009$ | $-0.760 \pm 0.195$ | $0.434 \pm 0.012$ | $-1.000 \pm 0.000$ |
| | Conformity | $\mathbf{0.672 \pm 0.000}$ | $\mathbf{1.000 \pm 0.000}$ | $0.623 \pm 0.000$ | $\mathbf{1.000 \pm 0.000}$ | $\mathbf{0.697 \pm 0.000}$ | $\mathbf{1.000 \pm 0.000}$ |
| | REPVLM | $0.656 \pm 0.002$ | $\mathbf{1.000 \pm 0.000}$ | $0.646 \pm 0.003$ | $\mathbf{1.000 \pm 0.000}$ | $0.618 \pm 0.002$ | $\mathbf{1.000 \pm 0.000}$ |

**Robustness Under Distribution Shift** On the shift benchmarks (ObjectNet, ImageNet-R, ImageNet-Sketch), the spread between methods widens. MCDO becomes high-variance on ObjectNet, GMM collapses on ImageNet-Sketch ($S \in [-1.000, -0.760]$), and Embedding Norm's Spearman drops to $0.430$ on ImageNet-R. ProbVLM is the most unstable proxy under the noisy LAION dataset, with Spearman near zero and very large variance under shift (e.g. $S = -0.052 \pm 0.809$ on ObjectNet, $S = 0.035 \pm 0.871$ on ImageNet-Sketch). Conformity is the most stable baseline, retaining $S = 1.000$ on most shift cells and edging out REPVLM on Acc@90 in some cells (notably ObjectNet/LAION and ImageNet-Sketch); W-CLIP is competitive elsewhere but its Spearman degrades on ImageNet-R (down to $0.842$). REPVLM is the only method to achieve $S = 1.000$ across *all nine* shift cells, leads on Acc@90 throughout the ImageNet-R column, and remains competitive on ObjectNet, including under the noisy LAION proxy.

**Density-estimation quality explains the ranking** Aggregating across the full grid, the new baselines order roughly as REPVLM > W-CLIP ≈ Conformity > GMM, with Embedding Norm, ProbVLM and MCDO orthogonal to this ranking as they perturb the backbone rather than estimate density. The ordering tracks the geometric awareness and density-estimation quality of each method: Riemannian non-parametric (REPVLM), Euclidean Gaussian on whitened embeddings (W-CLIP), vMF approximation (Conformity), and Euclidean Gaussian mixture (GMM). While individual density-aware baselines can match REPVLM on one metric

in specific cells, REPVLM is the only method that is consistently best or near-best on *both* metrics across in-distribution and distribution-shifted benchmarks. This empirical ordering directly supports the framing in Section 3: on-manifold embedding density is the operative quantity for epistemic uncertainty in pre-trained VLMs, and methods more faithful to this quantity yield more reliable estimates.

## 5.3. Ablation study

We conduct ablation studies on ImageNet-1K with proxy dataset Conceptual Captions to isolate the impact of our design choices on uncertainty estimation quality.

**Impact of Riemannian Geometry**   The defining characteristic of REPVLM is its adherence to the manifold geometry of the embedding space. We compare our Riemannian approach against two Euclidean variants: one using a uniform base distribution on the hypersphere and another using a standard Gaussian base. As shown in Figure 3 (left), the Riemannian formulation consistently outperforms both Euclidean variants across *all* rejection thresholds. This performance gain stems from respecting the intrinsic manifold structure of the embedding space. Since VLM embeddings are $\ell_2$-normalized and reside on the hypersphere $\mathbb{S}^{d-1}$, geodesic interpolation provides a more natural and accurate path than straight-line Euclidean paths that 'cut through' the sphere's interior.

**Scaling with Proxy Data**   We analyze the sensitivity of REPVLM to the volume of training data by varying the proxy dataset size from 50K to 1M image-caption pairs. As shown in Figure 3 (center), uncertainty quality improves as the size of the proxy dataset increases, allowing REPVLM to capture the empirical embedding distribution more accurately. However, the gains in accuracy at the 90% rejection threshold begin to plateau beyond 500K samples. This suggests that moderate-sized proxy datasets are sufficient for generating reliable epistemic uncertainty estimates, making REPVLM practical even with limited resources.

**ODE Integration Dynamics**   To evaluate the impact of numerical integration on uncertainty quality, we varied the number of ODE integration steps. As shown in Figure 3 (right), the results indicate that a single step is insufficient and two steps yield unstable results with high variance. Performance robustly converges at only five steps. This rapid stabilization is enabled by our manifold-aware integration strategy, which uses tangent space projections and renormalization to prevent numerical drift away from $\mathbb{S}^{d-1}$. Further, because the pre-trained CLIP embeddings are well-regularized within a highly structured joint semantic space, the learned vector field is sufficiently smooth to allow for rapid convergence. This ensures minimal additional over-

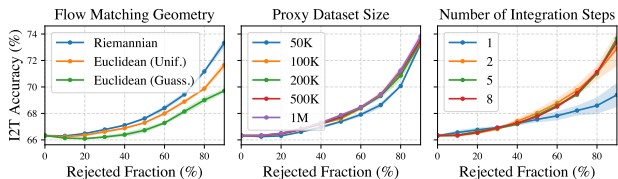

*Figure 3.* **Ablation studies of REPVLM. Left:** Comparison of the proposed Riemannian formulation against Euclidean variants using uniform and Gaussian base distributions. **Center:** Impact of proxy training data size, varying from 50K to 1M image-caption samples. **Right:** Convergence of accuracy-rejection performance relative to the number of ODE integration steps.

head, maintaining the inherent efficiency of REPVLM.

## 5.4. Computational Cost

Uncertainty estimation overhead is primarily dictated by the backbone model. While MCDO is the most computationally demanding approach (92.50 GFLOPs) due to $N = 10$ full forward passes, ProbVLM and REPVLM achieve significant gains by decoupling estimation from the heavy backbone. Although ProbVLM is efficient (10.35 GFLOPs), it yields unreliable epistemic uncertainty estimates, often showing poor or even negative correlation with model error. Leveraging a network architecture designed to match the unit cost of ProbVLM ($G_{\text{ProbVLM}} = G_{\text{REPVLM}} = 0.11$), REPVLM achieves even greater efficiency at 9.80 GFLOPs, requiring only 0.55 GFLOPs of additional overhead.

*Table 3.* Computational complexity comparison in GFLOPs. $N$ denotes the number of stochastic samples for MCDO and ProbVLM, while $T$ denotes the ODE integration steps in REPVLM.

| METHOD | COMPONENT | UNIT COST | H. PARAM. | GFLOPS |
|---|---|---|---|---|
| Backbone | $G_{\text{bkb}}$ | 9.25 | - | 9.25 |
| MCDO | $NG_{\text{bkb}}$ | 9.25 | $N = 10$ | 92.50 |
| ProbVLM | $G_{\text{bkb}} + NG_{\text{ProbVLM}}$ | 0.11 | $N = 10$ | 10.35 |
| REPVLM | $G_{\text{bkb}} + TG_{\text{REPVLM}}$ | 0.11 | $T = 5$ | 9.80 |

## 5.5. Applications

We now demonstrate the practical utility of REPVLM's uncertainty score beyond zero-shot classification. We showcase its effectiveness in identifying OOD data and performing automated data curation.

**Out-of-Distribution Detection**   Our uncertainty estimates provide a principled and natural mechanism for OOD detection by identifying samples that reside in low-density regions of the embedding manifold. We designate ImageNet-1K as the in-distribution (ID) dataset, with ImageNet-R serving as near-OOD and EuroSAT as far-OOD. Following our theoretical framework, we use the conditional negative log-likelihood of the visual modality as the scoring func-

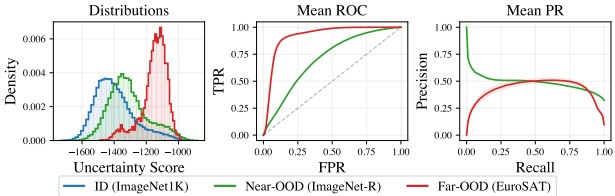

Figure 4. **Out-of-Distribution Detection. Left:** Distributions of uncertainty scores $U_{\text{ep}}(x) = -\log p(z|c)$ for ImageNet-1K (ID), ImageNet-R (Near-OOD), and EuroSAT (Far-OOD). **Center & Right:** Mean Receiver Operating Characteristic (ROC) and Mean Precision-Recall (PR) curves.

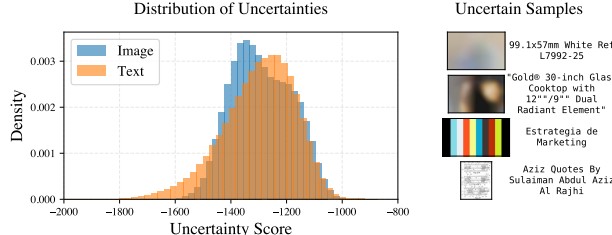

Figure 5. **Data Curation via Epistemic Uncertainty. Left:** The density distribution of uncertainty scores, where the tail corresponds to high-uncertainty samples. **Right:** Qualitative examples of samples with the highest epistemic uncertainty.

tion. As illustrated in Figure 4, standard ID inputs map to high-density regions, whereas OOD inputs consistently reside in the low-density regions. The structural separation between ID and OOD data results in robust detection performance, quantitatively supported by high Mean ROC and Mean Precision-Recall (PR) curves.

**Data Curation** The estimated epistemic uncertainty serves as an effective metric for cleaning large-scale datasets by identifying outliers and poor-quality samples. As illustrated in Figure 5, samples flagged with the highest uncertainty scores exhibit clear qualitative issues, such as heavy blurs or nonsensical visual and textual content. Filtering out these high-uncertainty samples will allow us to automatically prune noise from large-scale web datasets before training. The curation process potentially improves the stability and performance of future models without the need of expensive manual verification.

## 6. Conclusion and limitations

We introduced REPVLM, a principled framework for quantifying epistemic uncertainty in pre-trained Vision-Language Models using Conditional Riemannian Flow Matching. REPVLM estimates the probability density directly on the embedding manifold $\mathbb{S}^{d-1}$, establishing a rigorous link between embedding density and model confidence. Extensive experiments demonstrate that this density-based approach

provides a scalable and reliable metric for model confidence, significantly outperforming existing baselines in selective classification and out-of-distribution detection.

**Limitations** REPVLM has the following limitations: (I) The chain in Section 3 linking Bayesian epistemic uncertainty to embedding density is a motivating proxy argument, not a formal identity. A rigorous density-preservation result for non-invertible contrastive encoders remains open. (II) Like all post-hoc UQ methods, REPVLM requires a proxy dataset representative of the target VLM's pre-training domain. Specialised VLMs need in-domain proxies. (III) The theory assumes $\ell_2$-normalised embeddings on $\mathbb{S}^{d-1}$ from cosine-similarity-trained encoders and results should not be over-extrapolated to VLMs with different geometries. (IV) Density conflates rarity with model ignorance, posing fairness risks for underrepresented groups in automated curation. Per-subgroup evaluation or human review is recommended in safety- or fairness-critical settings.

## Impact Statement

This paper presents work whose goal is to advance the field of Machine Learning. One consequence worth flagging is that density-based uncertainty conflates rarity with model ignorance, so rare but valid inputs (e.g., underrepresented groups) may be flagged alongside genuine outliers. In fairness- or safety-critical use, high-uncertainty samples should be treated as candidates for human review rather than automatic removal.

## Acknowledgments

The computations/data handling were enabled by the Alvis resource provided by Alice Wallenberg Foundation at the National Supercomputer Centre and by the National Academic Infrastructure for Supercomputing in Sweden (NAISS) through project NAISS 2025/22-1350. LJ acknowledges funding from the Centre for Interdisciplinary Mathematics at Uppsala University. AH and PS acknowledge support from the Swedish Research Council through grant agreement nos. 2023-05167 and 2023-05593 respectively. EV and PS acknowledge support from the Wallenberg AI, Autonomous Systems and Software Program (WASP) funded by the Knut and Alice Wallenberg Foundation. EV additionally acknowledges support from the Kjell & Märta Beijer Foundation, and PS from the Data-Driven Life Science (DDLS) NEST project "TIMED".

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

# A. Derivations for Riemannian Flow Matching

In this section, we provide the complete mathematical derivations for our Conditional Riemannian Flow Matching framework on the hypersphere $\mathbb{S}^{d-1}$.

## A.1. Preliminaries: Geometry of the Hypersphere

The $(d-1)$-dimensional unit hypersphere embedded in $\mathbb{R}^d$ is defined as:

$$\mathbb{S}^{d-1} = \{z \in \mathbb{R}^d : \|z\|_2 = 1\}. \tag{1}$$

**Tangent Space.** The tangent space at a point $z \in \mathbb{S}^{d-1}$ is the set of all vectors orthogonal to $z$:

$$T_z\mathbb{S}^{d-1} = \{v \in \mathbb{R}^d : \langle v, z \rangle = 0\}. \tag{2}$$

This is a $(d-1)$-dimensional linear subspace of $\mathbb{R}^d$.

**Riemannian Metric.** The hypersphere inherits the standard Euclidean inner product as its Riemannian metric. For tangent vectors $u, v \in T_z\mathbb{S}^{d-1}$:

$$\langle u, v \rangle_z = u^\top v. \tag{3}$$

**Tangent Space Projection.** Given an arbitrary vector $v \in \mathbb{R}^d$, its projection onto the tangent space $T_z\mathbb{S}^{d-1}$ is:

$$\Pi_{T_z}(v) = v - \langle v, z \rangle z = (I - zz^\top)v. \tag{4}$$

This projection removes the component of $v$ normal to the sphere at $z$.

**Exponential Map.** The exponential map $\exp_z : T_z\mathbb{S}^{d-1} \to \mathbb{S}^{d-1}$ maps a tangent vector to a point on the sphere by following the geodesic:

$$\exp_z(v) = \cos(\|v\|)z + \sin(\|v\|)\frac{v}{\|v\|}. \tag{5}$$

For $v = 0$, we have $\exp_z(0) = z$.

**Geodesic Distance.** The geodesic (great-circle) distance between two points $z_0, z_1 \in \mathbb{S}^{d-1}$ is:

$$d_{\mathbb{S}}(z_0, z_1) = \arccos(\langle z_0, z_1 \rangle). \tag{6}$$

## A.2. Construction of Probability Paths

Flow matching constructs a time-dependent probability density $p_t(z)$ that interpolates between a simple base distribution $p_0$ and the target data distribution $p_1$.

**Base Distribution.** We use the uniform distribution on the hypersphere as our base:

$$p_0(z) = \mathrm{Unif}(\mathbb{S}^{d-1}) = \frac{1}{\mathrm{Vol}(\mathbb{S}^{d-1})}, \tag{7}$$

where the volume of the $(d-1)$-sphere is:

$$\mathrm{Vol}(\mathbb{S}^{d-1}) = \frac{2\pi^{d/2}}{\Gamma(d/2)}. \tag{8}$$

Sampling from this distribution can be achieved by drawing $\xi \sim \mathcal{N}(0, I_d)$ and normalizing: $z_0 = \xi/\|\xi\|$.

**Target Distribution.** The target distribution $p_1(z|c)$ is the empirical distribution of VLM embeddings for modality $c$:

$$p_1(z|c) = \frac{1}{N_c}\sum_{i=1}^{N_c} \delta(z - z_i^{(c)}), \tag{9}$$

where $\{z_i^{(c)}\}_{i=1}^{N_c}$ are the embeddings from the proxy dataset for modality $c$.

**Conditional Probability Path.** Following the Riemannian Flow Matching framework (Chen & Lipman, 2024), we construct a conditional probability path $p_t(z|z_1)$ that concentrates mass along the geodesic from a random base point $z_0$ to a fixed target point $z_1$. Marginalizing over the base distribution yields:

$$p_t(z) = \int_{\mathbb{S}^{d-1}} p_t(z|z_1)p_1(z_1)\,d\mu(z_1), \tag{10}$$

where $d\mu$ is the uniform measure on $\mathbb{S}^{d-1}$.

**Spherical Linear Interpolation (Slerp).** Given two points $z_0, z_1 \in \mathbb{S}^{d-1}$ with geodesic distance $\theta = \arccos(\langle z_0, z_1\rangle)$, the geodesic path connecting them is parameterized as:

$$z_t = \text{slerp}(z_0, z_1; t) = \frac{\sin((1-t)\theta)}{\sin\theta}z_0 + \frac{\sin(t\theta)}{\sin\theta}z_1, \tag{11}$$

for $t \in [0, 1]$. This satisfies the boundary conditions $z_0 = \text{slerp}(z_0, z_1; 0)$ and $z_1 = \text{slerp}(z_0, z_1; 1)$.

### A.3. Target Vector Field

The target (conditional) vector field $u_t(z_t|z_1)$ is defined as the velocity of the geodesic path at time $t$:

$$u_t(z_t|z_1) = \frac{dz_t}{dt}. \tag{12}$$

**Derivation.** Taking the derivative of the slerp formula:

$$\frac{dz_t}{dt} = \frac{d}{dt}\left[\frac{\sin((1-t)\theta)}{\sin\theta}z_0 + \frac{\sin(t\theta)}{\sin\theta}z_1\right] \tag{13}$$

$$= \frac{-\theta\cos((1-t)\theta)}{\sin\theta}z_0 + \frac{\theta\cos(t\theta)}{\sin\theta}z_1. \tag{14}$$

Simplifying:

$$u_t(z_t|z_1) = \frac{\theta}{\sin\theta}\left[\cos(t\theta)z_1 - \cos((1-t)\theta)z_0\right]. \tag{15}$$

### A.4. Loss Function

**Conditional Flow Matching Objective.** The flow matching objective trains a neural network $v_t(z, c; \phi)$ to approximate the marginal vector field by regressing onto the conditional target vector field:

$$\mathcal{L}_{\text{CFM}}(\phi) = \mathbb{E}_{t, z_0, z_1, c}\left[\|v_t(z_t, c; \phi) - u_t(z_t|z_1)\|^2\right], \tag{16}$$

where:

- $t \sim \text{Unif}([0, 1])$,

- $c \sim \text{Bernoulli}(0.5)$ (modality indicator),

- $z_1 \sim p_1(z|c)$ (target sample from proxy dataset),

- $z_0 \sim p_0 = \text{Unif}(\mathbb{S}^{d-1})$ (base sample),

- $z_t = \text{slerp}(z_0, z_1; t)$.

**Tangent Space Projection.** To ensure the learned vector field respects the manifold constraint, we apply tangent space projection to the network output:

$$v_t(z, c; \phi) = \Pi_{T_z}(\tilde{v}_t(z, c; \phi)) = \tilde{v}_t - \langle\tilde{v}_t, z\rangle z, \tag{17}$$

where $\tilde{v}_t$ is the raw network output.

---

**Algorithm 1** Training Conditional Riemannian Flow Matching

---

**Require:** Proxy dataset $\mathcal{D} = \{(z_i^{\text{img}}, z_i^{\text{txt}})\}_{i=1}^N$, network $v_\phi$
1: **while** not converged **do**
2:     Sample batch of pairs $(z^{\text{img}}, z^{\text{txt}}) \sim \mathcal{D}$ {Draw from proxy modality pairs}
3:     Sample $c \sim \text{Bernoulli}(0.5)$ for each pair {Select conditioning modality}
4:     Set $z_1 = z^{\text{img}}$ if $c = 0$, else $z_1 = z^{\text{txt}}$ {Define target sample}
5:     Sample $z_0 \sim \text{Unif}(\mathbb{S}^{d-1})$ {Sample from simple base distribution}
6:     Sample $t \sim \text{Unif}([0,1])$ {Draw random time step}
7:     Compute $\theta = \arccos(\langle z_0, z_1 \rangle)$ {Calculate geodesic distance}
8:     Compute $z_t = \frac{\sin((1-t)\theta)}{\sin\theta} z_0 + \frac{\sin(t\theta)}{\sin\theta} z_1$ {Spherical linear interpolation }
9:     Compute $u_t = \frac{\theta}{\sin\theta}[\cos(t\theta)z_1 - \cos((1-t)\theta)z_0]$ {Target geodesic velocity}
10:    Compute $\tilde{v}_t = v_\phi(z_t, t, c)$ {Evaluate raw network output}
11:    Project: $v_t = \tilde{v}_t - \langle \tilde{v}_t, z_t \rangle z_t$ {Project to tangent space $T_{z_t}\mathbb{S}^{d-1}$}
12:    Compute loss: $\mathcal{L} = \|v_t - u_t\|^2$ {Riemannian Flow Matching objective}
13:    Update $\phi$ via gradient descent {Optimize vector field parameters}
14: **end while**

---

**Training Algorithm.** The complete training procedure is summarized in Algorithm 1.

### A.5. Likelihood Computation

**Change of Variables on Manifolds.** For a flow $\psi_t : \mathbb{S}^{d-1} \to \mathbb{S}^{d-1}$ generated by the vector field $v_t$, the density evolves according to the continuity equation:

$$\frac{\partial p_t}{\partial t} + \text{div}_{\mathbb{S}^{d-1}}(p_t v_t) = 0. \tag{18}$$

This yields the instantaneous change of variables formula:

$$\frac{d \log p_t(z_t)}{dt} = -\text{div}_{\mathbb{S}^{d-1}}(v_t(z_t)). \tag{19}$$

**Log-Likelihood Computation.** Integrating from $t = 0$ to $t = 1$:

$$\log p_1(z_1|c) = \log p_0(z_0) - \int_0^1 \text{div}_{\mathbb{S}^{d-1}}(v_t(z_t, c; \phi)) \, dt, \tag{20}$$

where $z_0 = \psi_0(z_1)$ is obtained by integrating the ODE backward from $z_1$.

Since the base distribution is uniform:

$$\log p_0(z_0) = -\log \text{Vol}(\mathbb{S}^{d-1}) = -\log \frac{2\pi^{d/2}}{\Gamma(d/2)}. \tag{21}$$

**Riemannian Divergence.** The divergence of a vector field on $\mathbb{S}^{d-1}$ can be computed as:

$$\text{div}_{\mathbb{S}^{d-1}}(v) = \text{div}_{\mathbb{R}^d}(v) - (d-1)\langle v, z \rangle. \tag{22}$$

Since $v_t \in T_z\mathbb{S}^{d-1}$ implies $\langle v_t, z \rangle = 0$, this simplifies to:

$$\text{div}_{\mathbb{S}^{d-1}}(v_t) = \text{div}_{\mathbb{R}^d}(v_t) = \text{Tr}\left(\frac{\partial v_t}{\partial z}\right). \tag{23}$$

**Hutchinson Trace Estimator.** Computing the exact trace requires $\mathcal{O}(d)$ backward passes through the network. Instead, we use the Hutchinson estimator:

$$\text{Tr}(A) = \mathbb{E}_{\epsilon \sim p(\epsilon)}[\epsilon^\top A \epsilon], \tag{24}$$

where $p(\epsilon)$ is any distribution with $\mathbb{E}[\epsilon] = 0$ and $\text{Cov}(\epsilon) = I$.

For the manifold setting, we project the probe vector onto the tangent space:

$$\text{div}_{\mathbb{S}^{d-1}}(v_t) \approx \langle \tilde{\epsilon}, \nabla_z (v_t \cdot \tilde{\epsilon}) \rangle, \tag{25}$$

where $\tilde{\epsilon} = \Pi_{T_z}(\epsilon) = \epsilon - \langle \epsilon, z \rangle z$ and $\epsilon \sim \mathcal{N}(0, I_d)$.

**Numerical Integration.** We solve the reverse ODE using a Riemannian-adapted Euler's method. At each step, we:

1. Evaluate the vector field at intermediate points.

2. Project velocity evaluations onto the tangent space.

3. Update the position and renormalize to the sphere.

The complete inference algorithm is given in Algorithm 2.

---

**Algorithm 2** Uncertainty Inference via Likelihood Computation

---

**Require:** Query embedding $z_1 \in \mathbb{S}^{d-1}$, modality $c$, trained network $v_\phi$, steps $K$
1: Initialize $z \leftarrow z_1$, $\log p \leftarrow 0$, $\Delta t \leftarrow 1/K$
2: **for** $k = K, K-1, \ldots, 1$ **do**
3:    $t \leftarrow k/K$
4:    Sample $\epsilon \sim \mathcal{N}(0, I_d)$
5:    $\tilde{\epsilon} \leftarrow \epsilon - \langle \epsilon, z \rangle z$ {Project probe to tangent space $T_z \mathbb{S}^{d-1}$}
6:    $v \leftarrow v_\phi(z, t, c)$ {Evaluate learned vector field}
7:    $\text{div} \leftarrow \langle \tilde{\epsilon}, \nabla_z (v \cdot \tilde{\epsilon}) \rangle$ {Hutchinson trace estimator}
8:    $\log p \leftarrow \log p + \text{div} \cdot \Delta t$ {Integrate divergence term}
9:    $z \leftarrow z - v \cdot \Delta t$ {Reverse Euler step}
10:    $z \leftarrow z/\|z\|$ {Project back to sphere (renormalization)}
11: **end for**
12: $\log p_0 \leftarrow -\log(2\pi^{d/2}/\Gamma(d/2))$ {Uniform log-density on $\mathbb{S}^{d-1}$}
13: **return** $U_{\text{ep}} = -(\log p_0 - \log p)$ {Negative log-likelihood score}

---

## B. Experimental Details

### B.1. Training Details

**Proxy Datasets.** We train our flow matching models on three large-scale image-caption datasets:

- **Conceptual Captions (CC3M):** 1M randomly sampled image-caption pairs from the 3.3M dataset (Sharma et al., 2018).

- **DataComp-1B:** 1M randomly sampled pairs from the curated DataComp-1B pool (Gadre et al., 2023).

- **LAION-2B:** 1M randomly sampled pairs from LAION-2B-en (Schuhmann et al., 2022).

Results reported in Table 2 and Figure 2 are based on models trained with 500K pairs. All images are preprocessed using the standard preprocessing pipeline corresponding to the pre-trained VLM.

**Hyperparameters.** The complete hyperparameter configuration is provided in Table 4. The learning rate is chosen based on a grid search over the set $\{1 \times 10^{-6}, 5 \times 10^{-6}, 1 \times 10^{-5}, 5 \times 10^{-5}, 1 \times 10^{-4}\}$.

**Computational Resources.** All experiments were conducted on NVIDIA A100 GPUs (40GB).

*Table 4.* Training hyperparameters.

| Hyperparameter | REPVLM | ProbVLM |
|---|---|---|
| Optimizer | AdamW | AdamW |
| Learning rate | $1 \times 10^{-5}$ | $1 \times 10^{-4}$ |
| Weight decay | $1 \times 10^{-5}$ | $1 \times 10^{-5}$ |
| Batch size | 2048 | 2048 |
| Training steps | 400,000 | 400,000 |
| Learning rate schedule | Linear Warmup + Constant | Cosine Annealing |
| Warmup steps | 1,000 | 1,000 |

**VLM Backbones.** We use pre-trained and frozen VLM encoders from the Huggingface `transformers` library (Wolf et al., 2020):

- **CLIP ViT-B/32:** 512-dimensional embeddings, trained on LAION-2B (Cherti et al., 2023).

- **SigLIP ViT-B/16:** 768-dimensional embeddings, trained with sigmoid loss (Zhai et al., 2023).

## B.2. Model Architecture

The vector field network $v_\phi(z, t, c)$ is implemented as a deep residual network consisting of an input projection, a series of residual blocks with adaptive normalization, and a final output projection.

**Conditioning and Embedding.** The network processes time $t$ and modality $c$ into a shared conditioning vector:

- **Time $t$:** Encoded via sinusoidal positional encoding $\gamma(t)$ with 256 frequencies, followed by a 2-layer MLP to produce a $d_{\text{hidden}}$-dimensional embedding.

- **Modality $c$:** Embedded via a learnable lookup table into a $d_{\text{hidden}}$-dimensional vector.

- **Combined Context:** The conditioning vector $\mathbf{c}$ is formed by the element-wise sum of the time and modality embeddings.

**Residual Blocks.** Each of the 6 residual blocks employs an Adaptive Layer Normalization (AdaLN) mechanism. Given the input to the $i$-th block $h_i$, the transformation is defined as:

$$h_{i+1} = h_i + \text{Linear}\left(\text{SiLU}\left(\text{Norm}(h_i) \cdot (1 + \text{scale}(\mathbf{c})) + \text{shift}, (\mathbf{c}))\right)\right) \tag{26}$$

where $\text{scale}(\cdot)$ and $\text{shift}(\cdot)$ are linear projections of the conditioning vector $\mathbf{c}$. This allows the time and modality information to dynamically modulate the hidden representations throughout the network depth.

*Table 5.* Vector field network architecture specifications.

| Component | Specification | Dimension / Value |
|---|---|---|
| Input Projection | Linear Layer | $d \rightarrow d_{\text{hidden}}$ |
| Conditioning Dim ($d_{\text{hidden}}$) | Time + Modality | $d_{\text{hidden}}$ |
| Residual Blocks | Depth | 6 |
| Activation | Non-linearity | SiLU |
| Normalization | Type | AdaLN |
| Output Projection | Linear (Zero-init) | $d_{\text{hidden}} \rightarrow d$ |

## B.3. Full Ablation Study Results

We provide complete numerical results for all ablation studies conducted on ImageNet-1K with the Conceptual Captions proxy dataset.

*Table 6.* Comparison of Riemannian and Euclidean flow matching variants.

| Method | Acc. @ 90% Rej. | Spearman's $S$ |
|---|---|---|
| Euclidean + Gaussian base | $0.697 \pm 0.002$ | $0.893 \pm 0.052$ |
| Euclidean + Uniform base | $0.716 \pm 0.003$ | $0.973 \pm 0.012$ |
| Riemannian (Ours) | $\mathbf{0.733 \pm 0.004}$ | $\mathbf{0.993 \pm 0.006}$ |

### B.3.1. RIEMANNIAN VS. EUCLIDEAN FORMULATION

The Euclidean variants operate in $\mathbb{R}^d$ and project points onto the sphere only during evaluation. The "Euclidean + Gaussian base" uses $\mathcal{N}(0, I)$ as the base distribution, while "Euclidean + Uniform base" samples from the uniform distribution on $\mathbb{S}^{d-1}$ but uses straight-line interpolation. The Riemannian formulation outperforms both variants by 3-5% in accuracy at 90% rejection, demonstrating the importance of respecting the manifold geometry.

### B.3.2. PROXY DATASET SIZE

*Table 7.* Effect of proxy dataset size on uncertainty estimation quality.

| Dataset Size | Acc. @ 90% Rej. | Spearman's $S$ |
|---|---|---|
| 50K | $0.733 \pm 0.002$ | $0.973 \pm 0.012$ |
| 100K | $0.733 \pm 0.002$ | $0.990 \pm 0.005$ |
| 200K | $0.735 \pm 0.002$ | $0.993 \pm 0.006$ |
| 500K | $0.734 \pm 0.004$ | $0.988 \pm 0.013$ |
| 1M | $\mathbf{0.738 \pm 0.003}$ | $\mathbf{0.998 \pm 0.005}$ |

Performance improves significantly when the size of proxy dataset is small, after which gains plateau. This suggests that moderate-sized proxy datasets are sufficient for reliable uncertainty estimation, making our method practical with limited computational resources.

### B.3.3. NUMBER OF ODE INTEGRATION STEPS

*Table 8.* Effect of ODE integration steps on uncertainty quality and inference time.

| Steps | Acc. @ 90% Rej. | Spearman's $S$ | GFLOPS |
|---|---|---|---|
| 1 | $0.694 \pm 0.009$ | $0.990 \pm 0.009$ | 0.11 |
| 2 | $0.729 \pm 0.013$ | $0.993 \pm 0.006$ | 0.22 |
| 5 | $\mathbf{0.737 \pm 0.002}$ | $\mathbf{1.000 \pm 0.000}$ | 0.55 |
| 8 | $0.735 \pm 0.002$ | $0.995 \pm 0.006$ | 0.88 |

Performance stabilizes at 5 integration steps, with negligible improvement beyond this point. The rapid convergence is facilitated by: (1) the smoothness of learned vector fields in the well-structured CLIP embedding space, and (2) our manifold-aware integration scheme that prevents drift from the hypersphere.

### B.4. Classification Protocol

All classification results reported in this paper follow the standard zero-shot protocol introduced with CLIP (Radford et al., 2021), applied identically across CLIP and SigLIP backbones. Given a downstream dataset with class set $\mathcal{Y} = \{y_1, \ldots, y_K\}$, each class label is embedded into a textual prompt of the form "`a photo of a [class name]`", which is passed through the frozen text encoder $f_\theta^{\text{txt}}$ to obtain $\ell_2$-normalized class embeddings $\{t_k\}_{k=1}^{K} \subset \mathbb{S}^{d-1}$. For a test image $x$, the image encoder $f_\theta^{\text{img}}$ produces an $\ell_2$-normalized embedding $z = f_\theta^{\text{img}}(x) \in \mathbb{S}^{d-1}$, and the predicted class is selected by cosine-similarity argmax:

$$\hat{y}(x) = \underset{k \in \{1, \ldots, K\}}{\operatorname{argmax}} \langle z, t_k \rangle. \tag{27}$$

No classification head is trained on top of the backbone, and no labels from the evaluation datasets are used during training of either the VLM or REPVLM. The uncertainty score $U_{\text{ep}}(z)$ is used for ranking (selective classification, OOD detection, curation).

## B.5. Evaluation Datasets

*Table 9.* Summary of evaluation datasets.

| Dataset | Classes | Test Size | Description |
|---------|---------|-----------|-------------|
| ImageNet-1K | 1000 | 50,000 | Standard object classification |
| Food101 | 101 | 25,250 | Fine-grained food recognition |
| CIFAR-100 | 100 | 10,000 | Low-resolution ($32 \times 32$) images |
| ObjectNet | 313 | 50,000 | Objects in unusual poses/contexts |
| ImageNet-R | 200 | 30,000 | Artistic renditions of ImageNet classes |
| ImageNet-Sketch | 1000 | 50,889 | Sketch drawings of ImageNet classes |
| EuroSAT | 10 | 5,400 | Satellite imagery (far-OOD) |

## B.6. Extended Results on Other Backbones

To evaluate our method across different VLMs, we report results on `laion/CLIP-ViT-L-14-laion2B-s32B-b82K` and `timm/ViT-B-16-SigLIP2-256`.

**Discussion: scope of applicability across encoder objectives.** The CLIP ViT-L/14 results in Table 10 confirm the effectiveness of REPVLM on a larger CLIP backbone, with consistent gains over baselines across both in-distribution and distribution-shift benchmarks. The SigLIP ViT-B/16 results in Table 11 show a more nuanced picture: REPVLM maintains strong performance on distribution-shift benchmarks (ObjectNet, ImageNet-R, ImageNet-Sketch) but degrades on in-distribution benchmarks (ImageNet-1K, Food101, CIFAR-100) relative to its CLIP counterpart. We attribute this boundary to the assumptions underlying our theoretical motivation. REPVLM's guarantee requires $\ell_2$-normalized embeddings on $\mathbb{S}^{d-1}$ and an encoder geometry that is shaped primarily by semantic alignment: the (A3) link in Section 3 relies on alignment-and-uniformity dynamics that map semantically similar inputs to tight clusters and rare or OOD inputs to sparse regions. SigLIP departs from this regime: its multi-objective training (e.g., sigmoid contrastive loss combined with self-distillation and local feature extraction in subsequent variants) injects texture- and patch-level information into the embedding, so embedding density no longer purely reflects semantic coverage. For OOD inputs, which are unusual both semantically and texturally, density remains a reliable signal for semantics-based classification. For ID inputs, where errors arise from fine-grained class confusion rather than absence of training support, texture-driven density variation obscures the uncertainty–error correlation. This is analogous to how pixel-space generative models can fail at OOD detection by capturing low-level statistics rather than semantic content (Nalisnick et al., 2019). Despite this boundary, CLIP remains among the most widely deployed VL encoders in practice, making REPVLM applicable to an impactful class of models. We scope our claims accordingly and view the extension of the density-based framework to encoders trained with richer, multi-objective losses as a promising direction for future work.

*Table 10.* Evaluation results using CLIP ViT-L/14 backbone. We report Accuracy at 90% Rejection and Spearman's rank correlation $S$ between uncertainty and prediction error. The best results are highlighted in **bold**.

| EVAL. ON | METHOD | CONCEPTUAL CAPTIONS | | DATACOMP | | LAION | |
|---|---|---|---|---|---|---|---|
| | | Acc. @ 90% Rej. ↑ | $S$ ↑ | Acc. @ 90% Rej. ↑ | $S$ ↑ | Acc. @ 90% Rej. ↑ | $S$ ↑ |
| ImageNet-1K | MC-Dropout | 0.815 ± 0.006 | **1.000 ± 0.000** | 0.815 ± 0.006 | **1.000 ± 0.000** | 0.815 ± 0.006 | **1.000 ± 0.000** |
| | ProbVLM | 0.795 ± 0.005 | **1.000 ± 0.000** | 0.795 ± 0.012 | 0.998 ± 0.005 | 0.739 ± 0.013 | -0.232 ± 0.726 |
| | Emb. Norm | 0.845 ± 0.000 | **1.000 ± 0.000** | 0.845 ± 0.000 | **1.000 ± 0.000** | 0.845 ± 0.000 | **1.000 ± 0.000** |
| | W-CLIP | 0.778 ± 0.000 | 0.152 ± 0.000 | 0.778 ± 0.000 | 0.152 ± 0.000 | 0.778 ± 0.000 | 0.152 ± 0.000 |
| | GMM | 0.748 ± 0.013 | 0.503 ± 0.391 | 0.717 ± 0.006 | -0.942 ± 0.110 | 0.767 ± 0.008 | 0.833 ± 0.094 |
| | Conformity | 0.877 ± 0.000 | **1.000 ± 0.000** | 0.847 ± 0.000 | **1.000 ± 0.000** | 0.869 ± 0.000 | **1.000 ± 0.000** |
| | REPVLM | **0.885 ± 0.001** | **1.000 ± 0.000** | **0.865 ± 0.002** | **1.000 ± 0.000** | **0.877 ± 0.004** | **1.000 ± 0.000** |
| Food101 | MC-Dropout | 0.923 ± 0.015 | 0.615 ± 0.314 | 0.923 ± 0.015 | 0.615 ± 0.314 | 0.923 ± 0.015 | 0.615 ± 0.314 |
| | ProbVLM | 0.935 ± 0.004 | 0.990 ± 0.019 | 0.929 ± 0.009 | 0.726 ± 0.290 | 0.912 ± 0.009 | 0.135 ± 0.571 |
| | Emb. Norm | **0.988 ± 0.000** | **1.000 ± 0.000** | **0.988 ± 0.000** | **1.000 ± 0.000** | **0.988 ± 0.000** | **1.000 ± 0.000** |
| | W-CLIP | 0.921 ± 0.000 | 0.697 ± 0.000 | 0.921 ± 0.000 | 0.697 ± 0.000 | 0.921 ± 0.000 | 0.697 ± 0.000 |
| | GMM | 0.931 ± 0.007 | 0.983 ± 0.010 | 0.896 ± 0.005 | -0.627 ± 0.249 | 0.919 ± 0.007 | 0.932 ± 0.045 |
| | Conformity | 0.986 ± 0.000 | 0.988 ± 0.000 | 0.965 ± 0.000 | **1.000 ± 0.000** | **0.989 ± 0.000** | **1.000 ± 0.000** |
| | REPVLM | **0.988 ± 0.002** | **1.000 ± 0.000** | 0.969 ± 0.001 | **1.000 ± 0.000** | 0.986 ± 0.002 | **1.000 ± 0.000** |
| Cifar-100 | MC-Dropout | 0.898 ± 0.007 | 0.993 ± 0.006 | 0.898 ± 0.007 | 0.993 ± 0.006 | 0.898 ± 0.007 | 0.993 ± 0.006 |
| | ProbVLM | 0.887 ± 0.013 | **1.000 ± 0.000** | 0.861 ± 0.026 | 0.585 ± 0.561 | 0.863 ± 0.029 | 0.612 ± 0.705 |
| | Emb. Norm | 0.914 ± 0.000 | **1.000 ± 0.000** | 0.914 ± 0.000 | **1.000 ± 0.000** | 0.914 ± 0.000 | **1.000 ± 0.000** |
| | W-CLIP | 0.846 ± 0.000 | 0.212 ± 0.000 | 0.846 ± 0.000 | 0.212 ± 0.000 | 0.846 ± 0.000 | 0.212 ± 0.000 |
| | GMM | 0.885 ± 0.017 | 0.668 ± 0.203 | 0.801 ± 0.017 | -0.852 ± 0.164 | 0.837 ± 0.008 | 0.396 ± 0.427 |
| | Conformity | 0.963 ± 0.000 | **1.000 ± 0.000** | 0.925 ± 0.000 | **1.000 ± 0.000** | 0.967 ± 0.000 | **1.000 ± 0.000** |
| | REPVLM | **0.970 ± 0.003** | **1.000 ± 0.000** | **0.936 ± 0.003** | **1.000 ± 0.000** | 0.963 ± 0.006 | **1.000 ± 0.000** |
| ObjectNet | MC-Dropout | 0.775 ± 0.041 | 0.968 ± 0.057 | 0.775 ± 0.041 | 0.968 ± 0.057 | 0.775 ± 0.041 | 0.968 ± 0.057 |
| | ProbVLM | 0.677 ± 0.034 | 0.726 ± 0.426 | 0.587 ± 0.015 | -0.869 ± 0.095 | 0.621 ± 0.050 | -0.219 ± 0.806 |
| | Emb. Norm | 0.878 ± 0.000 | **1.000 ± 0.000** | **0.878 ± 0.000** | **1.000 ± 0.000** | **0.878 ± 0.000** | **1.000 ± 0.000** |
| | W-CLIP | 0.686 ± 0.000 | 0.903 ± 0.000 | 0.686 ± 0.000 | 0.903 ± 0.000 | 0.686 ± 0.000 | 0.903 ± 0.000 |
| | GMM | 0.606 ± 0.057 | -0.275 ± 0.752 | 0.717 ± 0.024 | 0.835 ± 0.090 | 0.739 ± 0.025 | 0.544 ± 0.103 |
| | Conformity | 0.871 ± 0.000 | **1.000 ± 0.000** | 0.826 ± 0.000 | **1.000 ± 0.000** | 0.839 ± 0.000 | **1.000 ± 0.000** |
| | REPVLM | **0.879 ± 0.004** | **1.000 ± 0.000** | 0.857 ± 0.003 | **1.000 ± 0.000** | 0.865 ± 0.005 | **1.000 ± 0.000** |
| ImageNet-R | MC-Dropout | 0.905 ± 0.020 | 0.884 ± 0.215 | 0.905 ± 0.020 | 0.884 ± 0.215 | 0.905 ± 0.020 | 0.884 ± 0.215 |
| | ProbVLM | 0.899 ± 0.004 | 0.990 ± 0.019 | 0.910 ± 0.014 | 0.998 ± 0.005 | 0.843 ± 0.034 | -0.534 ± 0.461 |
| | Emb. Norm | 0.962 ± 0.000 | **1.000 ± 0.000** | **0.962 ± 0.000** | **1.000 ± 0.000** | **0.962 ± 0.000** | **1.000 ± 0.000** |
| | W-CLIP | 0.853 ± 0.000 | -0.212 ± 0.000 | 0.853 ± 0.000 | -0.212 ± 0.000 | 0.853 ± 0.000 | -0.212 ± 0.000 |
| | GMM | 0.874 ± 0.006 | 0.712 ± 0.192 | 0.830 ± 0.005 | -0.809 ± 0.195 | 0.877 ± 0.006 | 0.386 ± 0.388 |
| | Conformity | 0.960 ± 0.000 | **1.000 ± 0.000** | 0.940 ± 0.000 | **1.000 ± 0.000** | 0.956 ± 0.000 | **1.000 ± 0.000** |
| | REPVLM | **0.965 ± 0.001** | **1.000 ± 0.000** | 0.957 ± 0.003 | **1.000 ± 0.000** | 0.960 ± 0.003 | **1.000 ± 0.000** |
| ImageNet-Sketch | MC-Dropout | 0.726 ± 0.014 | **1.000 ± 0.000** | 0.726 ± 0.014 | **1.000 ± 0.000** | 0.726 ± 0.014 | **1.000 ± 0.000** |
| | ProbVLM | 0.717 ± 0.013 | **1.000 ± 0.000** | 0.703 ± 0.021 | 0.983 ± 0.034 | 0.608 ± 0.015 | -0.537 ± 0.466 |
| | Emb. Norm | 0.824 ± 0.000 | **1.000 ± 0.000** | 0.824 ± 0.000 | **1.000 ± 0.000** | 0.824 ± 0.000 | **1.000 ± 0.000** |
| | W-CLIP | 0.532 ± 0.000 | -0.964 ± 0.000 | 0.532 ± 0.000 | -0.964 ± 0.000 | 0.532 ± 0.000 | -0.964 ± 0.000 |
| | GMM | 0.591 ± 0.010 | -0.828 ± 0.196 | 0.595 ± 0.015 | -0.566 ± 0.509 | 0.544 ± 0.020 | -1.000 ± 0.000 |
| | Conformity | 0.812 ± 0.000 | **1.000 ± 0.000** | 0.802 ± 0.000 | **1.000 ± 0.000** | 0.827 ± 0.000 | **1.000 ± 0.000** |
| | REPVLM | **0.855 ± 0.001** | **1.000 ± 0.000** | **0.840 ± 0.004** | **1.000 ± 0.000** | **0.836 ± 0.003** | **1.000 ± 0.000** |

*Table 11.* Evaluation results using SigLIP ViT-B/16 backbone. We report Accuracy at 90% Rejection and Spearman's rank correlation $S$ between uncertainty and prediction error. The best results are highlighted in **bold**.

| EVAL. ON | METHOD | CONCEPTUAL CAPTIONS | | DATACOMP | | LAION | |
|---|---|---|---|---|---|---|---|
| | | Acc. @ 90% Rej. ↑ | $S$ ↑ | Acc. @ 90% Rej. ↑ | $S$ ↑ | Acc. @ 90% Rej. ↑ | $S$ ↑ |
| ImageNet-1K | MC-Dropout | $0.772 \pm 0.014$ | $-0.770 \pm 0.248$ | $0.772 \pm 0.014$ | $-0.770 \pm 0.248$ | $0.772 \pm 0.014$ | $-0.770 \pm 0.248$ |
| | ProbVLM | $0.745 \pm 0.003$ | $-1.000 \pm 0.000$ | $0.802 \pm 0.010$ | $0.605 \pm 0.680$ | $0.749 \pm 0.004$ | $-1.000 \pm 0.000$ |
| | Emb. Norm | $\mathbf{0.863 \pm 0.000}$ | $\mathbf{1.000 \pm 0.000}$ | $\mathbf{0.863 \pm 0.000}$ | $\mathbf{1.000 \pm 0.000}$ | $\mathbf{0.863 \pm 0.000}$ | $\mathbf{1.000 \pm 0.000}$ |
| | W-CLIP | $0.687 \pm 0.000$ | $-1.000 \pm 0.000$ | $0.794 \pm 0.015$ | $-0.234 \pm 0.441$ | $0.680 \pm 0.000$ | $-1.000 \pm 0.000$ |
| | GMM | $0.688 \pm 0.006$ | $-1.000 \pm 0.000$ | $0.667 \pm 0.007$ | $-1.000 \pm 0.000$ | $0.672 \pm 0.008$ | $-1.000 \pm 0.000$ |
| | Conformity | $0.771 \pm 0.000$ | $-0.952 \pm 0.000$ | $\underline{0.818 \pm 0.000}$ | $\underline{0.964 \pm 0.000}$ | $\underline{0.802 \pm 0.000}$ | $\underline{0.939 \pm 0.000}$ |
| | REPVLM | $\underline{0.789 \pm 0.001}$ | $\underline{-0.693 \pm 0.006}$ | $\underline{0.779 \pm 0.001}$ | $0.573 \pm 0.000$ | $0.783 \pm 0.002$ | $-0.290 \pm 0.000$ |
| Food101 | MC-Dropout | $0.928 \pm 0.024$ | $0.164 \pm 0.230$ | $0.928 \pm 0.024$ | $0.164 \pm 0.230$ | $0.928 \pm 0.024$ | $0.164 \pm 0.230$ |
| | ProbVLM | $0.934 \pm 0.006$ | $0.358 \pm 0.684$ | $0.925 \pm 0.015$ | $-0.321 \pm 0.826$ | $0.926 \pm 0.009$ | $-0.463 \pm 0.623$ |
| | Emb. Norm | $\mathbf{0.977 \pm 0.000}$ | $\mathbf{1.000 \pm 0.000}$ | $\mathbf{0.977 \pm 0.000}$ | $\mathbf{1.000 \pm 0.000}$ | $\mathbf{0.977 \pm 0.000}$ | $\mathbf{1.000 \pm 0.000}$ |
| | W-CLIP | $0.885 \pm 0.000$ | $-1.000 \pm 0.000$ | $0.924 \pm 0.008$ | $-0.590 \pm 0.103$ | $0.879 \pm 0.001$ | $-1.000 \pm 0.000$ |
| | GMM | $0.885 \pm 0.005$ | $-1.000 \pm 0.000$ | $0.852 \pm 0.005$ | $-1.000 \pm 0.000$ | $0.878 \pm 0.004$ | $-1.000 \pm 0.000$ |
| | Conformity | $\underline{0.958 \pm 0.000}$ | $\mathbf{1.000 \pm 0.000}$ | $\underline{0.967 \pm 0.000}$ | $\mathbf{1.000 \pm 0.000}$ | $\underline{0.968 \pm 0.000}$ | $\mathbf{1.000 \pm 0.000}$ |
| | REPVLM | $0.951 \pm 0.006$ | $0.940 \pm 0.005$ | $0.938 \pm 0.003$ | $0.988 \pm 0.009$ | $0.960 \pm 0.006$ | $0.939 \pm 0.036$ |
| Cifar-100 | MC-Dropout | $0.747 \pm 0.017$ | $-0.137 \pm 0.531$ | $0.747 \pm 0.017$ | $-0.137 \pm 0.531$ | $0.747 \pm 0.017$ | $-0.137 \pm 0.531$ |
| | ProbVLM | $\mathbf{0.825 \pm 0.009}$ | $\mathbf{0.988 \pm 0.013}$ | $0.767 \pm 0.022$ | $0.118 \pm 0.519$ | $0.786 \pm 0.024$ | $0.678 \pm 0.436$ |
| | Emb. Norm | $0.725 \pm 0.000$ | $-0.588 \pm 0.000$ | $\underline{0.725 \pm 0.000}$ | $-0.588 \pm 0.000$ | $0.725 \pm 0.000$ | $-0.588 \pm 0.000$ |
| | W-CLIP | $0.792 \pm 0.001$ | $0.910 \pm 0.010$ | $0.681 \pm 0.050$ | $-0.159 \pm 0.410$ | $0.725 \pm 0.001$ | $-0.949 \pm 0.042$ |
| | GMM | $0.729 \pm 0.017$ | $-0.745 \pm 0.083$ | $0.634 \pm 0.025$ | $-0.998 \pm 0.005$ | $0.708 \pm 0.021$ | $-0.837 \pm 0.135$ |
| | Conformity | $\underline{0.796 \pm 0.000}$ | $\mathbf{0.988 \pm 0.000}$ | $\mathbf{0.853 \pm 0.000}$ | $\mathbf{1.000 \pm 0.000}$ | $\mathbf{0.809 \pm 0.000}$ | $\mathbf{1.000 \pm 0.000}$ |
| | REPVLM | $0.787 \pm 0.005$ | $0.920 \pm 0.049$ | $0.697 \pm 0.006$ | $\underline{0.863 \pm 0.000}$ | $\underline{0.768 \pm 0.007}$ | $\underline{0.790 \pm 0.000}$ |
| ObjectNet | MC-Dropout | $0.691 \pm 0.048$ | $0.537 \pm 0.290$ | $0.691 \pm 0.048$ | $0.537 \pm 0.290$ | $0.691 \pm 0.048$ | $0.537 \pm 0.290$ |
| | ProbVLM | $0.522 \pm 0.020$ | $-1.000 \pm 0.000$ | $0.533 \pm 0.013$ | $-1.000 \pm 0.000$ | $0.597 \pm 0.029$ | $-0.639 \pm 0.373$ |
| | Emb. Norm | $\underline{0.891 \pm 0.000}$ | $\mathbf{1.000 \pm 0.000}$ | $\underline{0.891 \pm 0.000}$ | $\mathbf{1.000 \pm 0.000}$ | $\mathbf{0.891 \pm 0.000}$ | $\mathbf{1.000 \pm 0.000}$ |
| | W-CLIP | $0.512 \pm 0.002$ | $-1.000 \pm 0.000$ | $0.609 \pm 0.024$ | $-0.353 \pm 0.592$ | $0.388 \pm 0.002$ | $-1.000 \pm 0.000$ |
| | GMM | $0.489 \pm 0.021$ | $-0.985 \pm 0.029$ | $0.639 \pm 0.015$ | $0.030 \pm 0.117$ | $0.577 \pm 0.026$ | $-0.670 \pm 0.155$ |
| | Conformity | $0.818 \pm 0.000$ | $\mathbf{1.000 \pm 0.000}$ | $0.822 \pm 0.000$ | $\mathbf{1.000 \pm 0.000}$ | $0.844 \pm 0.000$ | $\mathbf{1.000 \pm 0.000}$ |
| | REPVLM | $\mathbf{0.920 \pm 0.002}$ | $\mathbf{1.000 \pm 0.000}$ | $\mathbf{0.914 \pm 0.025}$ | $\mathbf{1.000 \pm 0.000}$ | $\underline{0.884 \pm 0.006}$ | $\mathbf{1.000 \pm 0.000}$ |
| ImageNet-R | MC-Dropout | $0.943 \pm 0.011$ | $0.935 \pm 0.053$ | $0.943 \pm 0.011$ | $0.935 \pm 0.053$ | $0.943 \pm 0.011$ | $0.935 \pm 0.053$ |
| | ProbVLM | $0.909 \pm 0.002$ | $0.586 \pm 0.373$ | $0.926 \pm 0.007$ | $0.959 \pm 0.077$ | $0.915 \pm 0.008$ | $0.654 \pm 0.574$ |
| | Emb. Norm | $\mathbf{0.971 \pm 0.000}$ | $\mathbf{1.000 \pm 0.000}$ | $\mathbf{0.971 \pm 0.000}$ | $\mathbf{1.000 \pm 0.000}$ | $\mathbf{0.971 \pm 0.000}$ | $\mathbf{1.000 \pm 0.000}$ |
| | W-CLIP | $0.877 \pm 0.001$ | $-1.000 \pm 0.000$ | $0.910 \pm 0.004$ | $0.464 \pm 0.320$ | $0.836 \pm 0.000$ | $-1.000 \pm 0.000$ |
| | GMM | $0.870 \pm 0.005$ | $-1.000 \pm 0.000$ | $0.806 \pm 0.009$ | $-1.000 \pm 0.000$ | $0.849 \pm 0.013$ | $-1.000 \pm 0.000$ |
| | Conformity | $0.920 \pm 0.000$ | $0.988 \pm 0.000$ | $0.951 \pm 0.000$ | $\mathbf{1.000 \pm 0.000}$ | $0.953 \pm 0.000$ | $\mathbf{1.000 \pm 0.000}$ |
| | REPVLM | $\underline{0.961 \pm 0.003}$ | $\underline{0.998 \pm 0.005}$ | $\underline{0.970 \pm 0.002}$ | $\mathbf{1.000 \pm 0.000}$ | $\mathbf{0.971 \pm 0.001}$ | $\mathbf{1.000 \pm 0.000}$ |
| ImageNet-Sketch | MC-Dropout | $0.728 \pm 0.021$ | $0.905 \pm 0.150$ | $0.728 \pm 0.021$ | $0.905 \pm 0.150$ | $0.728 \pm 0.021$ | $0.905 \pm 0.150$ |
| | ProbVLM | $0.692 \pm 0.011$ | $0.753 \pm 0.184$ | $0.704 \pm 0.011$ | $0.944 \pm 0.073$ | $0.674 \pm 0.015$ | $0.341 \pm 0.597$ |
| | Emb. Norm | $\underline{0.798 \pm 0.000}$ | $0.988 \pm 0.000$ | $\mathbf{0.798 \pm 0.000}$ | $\underline{0.988 \pm 0.000}$ | $\mathbf{0.798 \pm 0.000}$ | $0.988 \pm 0.000$ |
| | W-CLIP | $0.608 \pm 0.001$ | $-0.993 \pm 0.006$ | $0.621 \pm 0.004$ | $-0.944 \pm 0.015$ | $0.557 \pm 0.001$ | $-1.000 \pm 0.000$ |
| | GMM | $0.532 \pm 0.012$ | $-1.000 \pm 0.000$ | $0.531 \pm 0.007$ | $-1.000 \pm 0.000$ | $0.481 \pm 0.006$ | $-1.000 \pm 0.000$ |
| | Conformity | $0.699 \pm 0.000$ | $0.879 \pm 0.000$ | $0.743 \pm 0.000$ | $\underline{0.988 \pm 0.000}$ | $0.728 \pm 0.000$ | $\underline{0.988 \pm 0.000}$ |
| | REPVLM | $\mathbf{0.834 \pm 0.003}$ | $\mathbf{1.000 \pm 0.000}$ | $\underline{0.779 \pm 0.004}$ | $\mathbf{1.000 \pm 0.000}$ | $\underline{0.796 \pm 0.002}$ | $\mathbf{1.000 \pm 0.000}$ |

