# OpenReview forum: "Epistemic Uncertainty Quantification for Pre-trained VLMs via Riemannian Flow Matching"
_ICML.cc/2026/Conference — ICML 2026 regular_

### Official Review · Reviewer_S4Jo · 2026-03-03

**Soundness:** 2
**Presentation:** 3
**Significance:** 3
**Originality:** 2
**Overall Recommendation:** 4
**Confidence:** 4

**Summary:**

This paper proposes REPVLM to estimate epistemic uncertainty in pre-trained vision-language models (VLMs). It models the density of embeddings on the hypersphere and uses low density as a signal of high uncertainty. The proposed method does not require retraining the backbone.

**Compliance With Llm Reviewing Policy:**

Affirmed.

**Final Justification:**

I thank the authors for their clear and detailed rebuttal, which addressed my main concerns and improved the overall understanding of the paper.  The work is technically solid, and the idea of performing density estimation on the hypersphere using Riemannian flow matching is well aligned with the geometry of vision-language model embeddings. The rebuttal provided helpful clarification on the theoretical assumptions, computational cost, and training stability, making the approach more convincing in practice. In terms of significance, the method is practical and scalable, as it does not require retraining the backbone, and it offers a useful way to estimate epistemic uncertainty in large pre-trained models. While the core idea of density-based uncertainty is related to prior work, the proposed formulation on the embedding manifold provides a meaningful extension and avoids some known limitations of earlier approaches.

Some limitations remain, particularly regarding reliance on a representative proxy dataset and potential challenges under domain shift. However, these are now better acknowledged and do not undermine the overall contribution. Based on these points, I find the paper to be a well-motivated and useful contribution with solid empirical support. The rebuttal reinforced my initial assessment, and I maintain my recommendation of weak accept.

**Key Questions For Authors:**

I have the following major questions regarding the proposed REPVLM model:
1. How robust is REPVLM when the embedding distribution significantly deviates from the assumptions made by the flow model?
2. Can the method handle extreme out-of-distribution inputs or completely novel concepts not represented in the proxy dataset?
3. How sensitive is the performance to the choice and domain of the proxy dataset used for density learning?
4. How does the proposed REPVLM model compare computationally and empirically to strong uncertainty baselines (e.g., ensembles or MC-Dropout) at scale?
5. Are these assumptions, like local linearity, really valid for today’s large vision-language models?
Meanwhile, addressing the following minor question is also important:
6. What is the training cost of the Riemannian flow model relative to the backbone size?

**Limitations:**

The paper discusses some assumptions, but it does not fully address sensitivity to proxy data, failure cases under large domain shift, or risks of relying purely on embedding density in safety-critical settings. A clearer discussion of these limitations and potential societal risks would improve the work.

**Strengths And Weaknesses:**

In my view, the major strengths are as follows:
1. The proposed method is technically good. Modeling the embedding density directly on the hypersphere is a well-thought idea.
2. Using Riemannian Flow Matching to estimate density is interesting which matches well with normalized VLM embeddings.
3. Moreover, the proposed method does not need retraining or changing the backbone model. This makes it practical and easier to scale.
However, I see some weaknesses:
4. The theory depends on strong assumptions, like local linearity and near-isometry. These may not fully hold in large VLMs.
5. Density-based uncertainty may depend a lot on how well the flow model is trained. It can also become unstable in very high-dimensional spaces.
6. Training the flow model is not cheap. It requires more computation compared to simpler post-hoc methods.
7. Although the formulation is new, the main idea of using density for uncertainty is related to earlier likelihood-based methods.

---

> ### Author Rebuttal · Authors · 2026-03-30
>
> We thank the reviewer for the constructive feedback and recognition of the method's technical merit and practicality.
>
> **SW4 & Q5: Theoretical assumptions.**
>
> While the assumptions may seem strong, they are rooted in deep learning theory and empirically supported:
>
> *   **Local linearity** is a standard assumption in Bayesian DL and UQ (Equation 10 in [1]). The limitation of the assumption is that it only holds for a sharp parameter posterior. Because the VLMs we use are optimized over billions of training samples, converging to minima with a sharp posterior, the first-order approximation is valid in this context.
> *   **Near-isometry** was used to establish the link between data density and embedding density. Rather than assuming strict isometry (violated by dimensionality-reducing encoders), we justify this link through contrastive learning dynamics. VLMs optimized for *alignment and uniformity* on the hypersphere map high-density regions into tightly packed semantic clusters (high $p(z)$), while OOD inputs are isolated into sparse regions (low $p(z)$)  [2]. This surrogate relationship is empirically validated by recent works like W-CLIP [3]. The updated justification will be reflected in the manuscript.
>
> **SW5: Flow model training stability**
>
> Three factors mitigate instability of the training of flow matching models in our work:
>
> 1.  **Low dimensionality:** We model density in the embedding space, not raw pixel/text space; a 20.5M parameter network efficiently captures this for CLIP embeddings.
> 2.  **Geometric regularization:** The $\mathbb{S}^{d-1}$ geometry constraint structurally regularizes the flow.
> 3.  **Data Efficiency:** Our ablation (Figure 3) shows that a massive dataset is not required; performance plateaus at around 500K proxy samples.
>
> **SW6, Q4, & Q6: Computational cost vs. Baselines at scale**
>
> The computational overhead of REPVLM is minimal during both training and inference:
>
> *   **Training:** Training takes only ~4 hours on a A100 GPU (one-time) on frozen embeddings.
> *   **Inference**: As shown in Table 3, requiring only a few integration steps, REPVLM's inference overhead adds only 0.55 GFLOPs per sample. In contrast, MCDO requires 92.50 GFLOPs for 10 backbone passes, and Deep Ensembles are infeasible for foundation VLMs, as accessing to multiple checkpoints pre-trained under identical configurations with varying random seeds is impractical.
>
> **SW7: Relation to earlier likelihood-based methods**
>
> We will add relevant citations. The critical differentiator is that REPVLM performs manifold-native density estimation in _semantic embedding space_, avoiding the known failure mode where pixel-space generative models assign high likelihoods to OOD data due to low-level statistics rather than semantic content [4].
>
> **Q1: Deviations from Flow Model Assumptions**
>
> If referring to geometric constraints: our ablation (Figure 3) shows Euclidean variants degrade significantly. If referring to distributional complexity (multimodality, heavy tails): flow matching is a _nonparametric_ density estimator that learns arbitrary vector fields without assuming a parametric form, making it robust to complex distributional structure. Additional [**results**](https://anonymous.4open.science/r/icml26_rebuttal-00EF/clip.png) with new baselines confirms that parametric assumptions degrade performance, as discussed in response to Reviewer 2 (StS3) W2.
>
> **Q2: Handling extreme OOD inputs or novel concepts**
>
> REPVLM is designed for this: novel concepts absent from the proxy dataset map to sparse manifold regions and receive high epistemic uncertainty. Our OOD detection experiments (Figure 4) with EuroSAT (satellite imagery, far from typical web images) confirm clear separation.
>
> **Q3: Proxy data sensitivity & Domain shifts & Limitations**
>
> Because foundational VLMs are trained on web-scale data, general-purpose datasets (CC3M, DATACOMP, LAION) act as natural proxies for "general knowledge" uncertainty. Table 2 confirms consistent results across three distinct proxy datasets. For specialized VLMs, an in-domain proxy is required. We will emphasize proxy selection in Limitations.
>
> **References**
>
> [1] Immer, et al. Improving predictions of Bayesian neural nets via local linearization. *AISTATS*, 2021.
>
> [2] Wang T., and Isola P. Understanding contrastive representation learning through alignment and uniformity on the hypersphere. *ICML*, 2020.
>
> [3] Betser R., et al. Whitened CLIP as a Likelihood Surrogate of Images and Captions. *ICML*, 2025.
>
> [4] Nalisnick, E., et al. Do Deep Generative Models Know What They Don't Know? *ICLR*, 2019.

---

> > ### Author Rebuttal · Reviewer_S4Jo · 2026-04-03
> >
> > Thank you for your detailed and well-organized rebuttal. I appreciate the clear answers to my questions. Overall, my main concerns have been addressed. The explanation of the theoretical assumptions (like local linearity and near-isometry) is clearer now, especially with the connection to previous work and contrastive learning in VLMs. Even though these assumptions are still approximations, the added explanation and experiments make them more convincing.
> >
> > I also appreciate the clarification about training stability and computational cost. The comparison with MC-Dropout and the information about training and inference cost help show that the method can be practical at scale. The explanation of why flow matching is stable in the embedding space is helpful as well. The discussion about proxy data and OOD generalization is also reassuring. It is good to see that the method works consistently across different proxy datasets, and that the limitations of choosing proxy data will be explained more clearly in the paper. Finally, the explanation of how this method is different from previous likelihood-based approaches makes the contribution clearer. Based on these explanations, I believe my main concerns are resolved, so I will keep my score the same.

---

> > > ### Author Response · Authors · 2026-04-03
> > >
> > > We sincerely thank the reviewer for acknowledging that the concerns have been resolved. We are glad that the clarifications on theoretical assumptions, computational efficiency, and proxy data sensitivity were helpful. We will incorporate the promised revisions including strengthened justification of near-isometry, additional citations to earlier likelihood-based methods, and a more detailed discussion of proxy data selection in the Limitations section.

---

### Official Review · Reviewer_e7px · 2026-03-07

**Soundness:** 3
**Presentation:** 2
**Significance:** 3
**Originality:** 3
**Overall Recommendation:** 4
**Confidence:** 4

**Summary:**

This paper proposes an epistemic uncertainty quantification method (REPVLM) based on embedding density estimation. REPVLM applies Riemannian Flow Matching on the spherical manifold to learn conditional embedding densities as the uncertainty score. Experimental results validate the effectiveness of this measure, especially the correlation between uncertainty and prediction errors, which looks near-perfect on some datasets.

**Compliance With Llm Reviewing Policy:**

Affirmed.

**Final Justification:**

Thanks to the authors for providing answers to my questions.

The authors' response to the core theoretical concern (Q1) remains at an intuitive level and fails to provide rigorous mathematical proof.

While the rebuttal has partially resolved my primary concerns, I stand by my original score.

**Key Questions For Authors:**

1. My most concern is the discussion of mechanism about Riemannian Flow Matching (manifold). Why does a transformation from Euclidean to Rieman manifold with different distance measures can have these advantages? Its deep mechanism is unclear. In other words, under what condition, we can make sure, with a high probability, the learning on Reiman manifold (the relationship between error and uncertainty) can be better (showing near-perfect) than on Euclidean space? Maybe this is a general difficult problem for all learning on Rieman manifold, but at least, the authors should have discussions by showing some viewpoints themselves.
2. Another is the rate of AI helping writing. To what extent did the authors use LLMs to assist with writing the manuscript? It should be clearly claimed. Some parts, e.g., the “Conclusion and limitation” seem as 100% AI-generated.

**Limitations:**

See my above comments and key questions.

**Strengths And Weaknesses:**

Strengths：
1.	The topic is worth discussion as epistemic uncertainty quantification is an important problem and has attracted increasing attention.
2.	The proposed method is relatively innovative. Compared with traditional estimation methods, REPVLM avoids parameter space sampling and achieves efficient density estimation by learning a vector field.
3.	The manuscript is well organized and written.

Weakness：
1. The theoretical analysis is insufficient. There is a gap between the definitions of epistemic uncertainty from the Bayesian perspectives $U_{ep}(x)=Tr(Cov_{p(\theta\mid D)}(z))$ and density perspectives $U_{ep}(x)=-\log p(z\mid c)$, as the manuscript does not provide a derivation for replacing $Tr(Cov(z))$ with $−\log p(z)$.
2. The comparison between density estimation methods in Euclidean space and REPVLM should be discussed, like the Gaussian mixture model (GMM), to demonstrate the benefits of hyper-spherical density modeling for VLMs and the superiority of RFM.
3. As mentioned in the limitations, density-based estimation may be unfair for tail class with few samples. This point should be stated explicitly in the introduction or the method section to clarify that the proposed quantification method is primarily applicable to balanced datasets.
4. There are two typos: 1. "p(x) ∝ p(z)" (in line 202) should be the same as "p(z)∝ p(x)" (in line 202); 2. the phrase “\ell-2 normalized embedding” should be “\ell_2 normalized embedding.”

---

> ### Author Rebuttal · Authors · 2026-03-30
>
> We sincerely thank the reviewer for the positive assessment, recognition of the method's novelty, and constructive suggestions.
>
> **W1: Gap between Bayesian and density definitions.**
>
> The Bayesian definition $U_\text{ep}(x) = \text{Tr}(\text{Cov}_{p(\theta|D)}(z))$ quantifies output spread under parameter uncertainty. Our claim is that $-\log p(z)$ serves as a _proxy_, not an identity. The chain in Section 3.1 establishes:
>
> 1. High epistemic uncertainty (Bayesian) $U_\text{ep}$ correlates with a large Parameter-Jacobian norm $\|J_\theta(x)\|_F$.
> 2. Empirical risk minimization forces the Jacobian norm to be small in regions of high data density $p(x)$.
> 3. High data density $p(x)$ and high latent density $p(z)$ are highly correlated.
>
> We will add an explicit remark: _"The relationship is established as a monotonic correspondence under the stated assumptions._ Also all assumptions in building this relation are detailed in the reponse to Reviewer 4 (S4Jo) under SW4 & Q5.
>
> **W2: Comparison with Euclidean density estimation method**
>
> The fundamental reason for modeling on the hypersphere is that modern VLMs trained via cosine-similarity-based loss (like CLIP and SigLIP) inherently project their representations onto an $\ell_2$-normalized unit hypersphere $\mathbb{S}^{d-1}$ [1]. Applying standard Euclidean density models (like a Gaussian Mixture Model) to these embeddings ignores this geometric constraint and wastes representational capacity outside the manifold. The results of suggested GMM baseline provided [**here**](https://anonymous.4open.science/r/icml26_rebuttal-00EF/clip.png) together with other additional baselines illustrate the failure of Euclidean parametric models in the hyperspherical spaces. Furthermore, our ablation (Figure 3, left) further shows 3–5% accuracy gains from the Riemannian formulation over Euclidean flow match variants.
>
> **W3: Unfairness to tail classes.**
>
> From a Uncertainty Quantification perspective, this is exactly correct behavior: epistemic uncertainty _should_ be high where the model has seen few examples. However, from a *downstream fairness* perspective, discarding high-uncertainty samples might disproportionately reject underrepresented demographic groups. Without side-information, distinguishing "rare but valid" from "noisy" is an inherent limitation of unsupervised density estimation, shared by all epistemic UQ methods. We will emphasize this in the camera-ready, recommending per-subgroup evaluation when metadata is available.
>
> **W4: Typos in the manuscript.**
>
> Thank you -- we will make corrections throughout.
>
> **Q1: Why Riemannian outperforms Euclidean-mechanism.**
>
> The core mechanism is _geometric consistency_. Because contrastive/sigmoid-based representation learning aligns embeddings by maximizing cosine similarity, the meaningful normalized embeddings live on the surface of the hypersphere $\mathbb{S}^{d-1}$. In Euclidean space, the model wastes capacity modeling the empty volume inside and outside the sphere, and interpolation paths of Euclidean flow matching cut straight *through* the interior. The model must implicitly learn to project these unnatural paths back onto the surface, requiring more parameters and data. By using Riemannian Flow Matching, we enforce the $\mathbb{S}^{d-1}$ constraint by design, ensuring vector fields flow along geodesics. The condition under which that guarantees better performance is when the underlying manifold is known (as VLM encoders explicitly enforce $z = f(x) / \|f(x)\|_2$). Baking this prior into the ODE results in a more data- and computation-efficient, and accurate vector field.
>
> **Q2: LLM-assistance with writing.**
>
> We used LLMs strictly for grammar and readability polishing, permitted under conference guidelines. All intellectual content, derivations, and experiments are the authors' own. We declared this in the submission checklist and will add an explicit statement in the camera-ready version.
>
> **References**
>
> [1] Wang T., and Isola P. Understanding contrastive representation learning through alignment and uniformity on the hypersphere. *ICML*, 2020.

---

> > ### Author Rebuttal · Reviewer_e7px · 2026-04-03
> >
> > Thank you for the clarification.
> >
> > The added results demonstrate the superiority of REPVLM. As for my main concern(Q1), while the geometric intuition is helpful, the explanation remains at an intuitive level and does not fully address my concern regarding the underlying mechanism.
> >
> > After careful consideration and reading other reviewers' comments, I would maintain my original rating for this paper.

---

> > > ### Author Response · Authors · 2026-04-03
> > >
> > > We thank the reviewer for the continued engagement. We would like to clarify the nature of our argument.
> > >
> > > VLMs trained with cosine-similarity objectives produce $\ell_2$-normalized embeddings on $\mathbb{S}^{d−1}$ by construction, and the Riemannian formulation simply respects this pre-existing constraint rather than introducing additional assumptions. From this perspective, Euclidean methods are the ones making an approximation, modeling density in flat $\mathbb{R}^d$ including regions where valid embeddings never exist (the hypersphere's interior and exterior), while Riemannian FM operates in the native space. The empirical gains ($3–5$% in Table 6; GMM and Euclidean baselines underperforming) then follow naturally: modeling in the correct geometry avoids wasting capacity on impossible regions and ensures interpolation paths remain on the manifold.
> > >
> > > We acknowledge that a full theoretical characterization of how much geometric mismatch costs in $d$ dimensions remains an open and interesting question; but we believe the combination of geometric reasoning and consistent empirical evidence provides substantive justification beyond intuition alone.

---

### Official Review · Reviewer_StS3 · 2026-03-07

**Soundness:** 3
**Presentation:** 3
**Significance:** 3
**Originality:** 2
**Overall Recommendation:** 4
**Confidence:** 5

**Summary:**

The paper proposing REPVLM, a Flow Matching based method that aim to use the density of VLMs latent space in order to estimate the confidence of the sample accuracy in VLMs.

**Compliance With Llm Reviewing Policy:**

Affirmed.

**Final Justification:**

The paper, as I said before, is on timely and highly relevant topic. The additional refs are more than necessary to be added. I think that the comparison against current measurements is important, and show on which cases your method is superior, and where it is less (interesting why). Upon additions, and clarifications my concerns are resolved.

**Key Questions For Authors:**

* What is your intuitive explanation regarding the good performance of your proposed method to domain shift? At the end, it was trained on a specific dataset, then what makes it generalizable in your sense?

* The details here are important, I couldn't find how do you obtain classification? If I think of CLIP then it is an embedder, I can think of two ways (maybe there are more) that one could classify: 1) pretrained classification head which was trained on top some given dataset. 2) open-vocabulary classification - namely find the best match against a predefined set of class prompts. So, how do you make classification in your case?


Overall, the direction and the techniques used are promising, the topic is highly relevant, nevertheless I feel like the paper missed several important papers in the field of representation learning in CLIP and in VLMs in general, where several measurements are already introduced in this field. Thus, I think that the authors should provide a correlation analysis to all these simple measurements to show if they caught something different. Also, I think that in terms of different VLMs, they provide too shallow analysis (two VLMs, single arch each) to state that their method is applicable for VLMs in general.

**Strengths And Weaknesses:**

Strengths:

* Analyzing the latent space of VLMs is of high importance for large amount of applications, moreover it is a highly timely topic.

* The link between density estimation and FM is interesting.

Weaknesses:

* The authors do not refer to several papers on this field which are also highly related [1,2,3,4], and should be referenced. Moreover, with respect to W-CLIP [1] that thoroughly analyzed the log likelihood derived from CLIP, this paper seems like another perspective of something that was already studied lately, maybe with different goal (Namely UQ). Moreover, they also showed how it can be used for OOD detection (with follow-up paper specifically on this topic [5]). Since you are working (as far as I understands it) for a single modality at each time, it is worth to mention also [6] (and several more), talking on the Modality Gap which seems relevant for this topic.

* The measurement provided by this paper should be compared against the followings: 1) log-likelihood - like in [1], conformity - like in [2], embedding norm - as in [3], and the density ratio - as in [4].

* The paper used the term VLMs throughout the paper, however they compare their method only on CLIP (single arch.) and SigLip (single  arch.), in my opinion it is too shallow analysis for stating that it holds for VLMs.

Minor Weaknesses:

* Line 32 right (and many other places), "... the hyper-spherical geometry of CLM embedding spaces.." - in [2] it has been shown that CLIP spanned in ellipsoid (some variances are much higher than others), so this statement is at least not accurate. Unless you are working with the normalized embeddings.

* It would be nice to add the most optimistic result on 90% rejection, to understand how close are the methods to this ceiling (say by filtering only falsely classified samples), because there is a large variance across different datasets. Moreover there is a pretty large variance for SigLip as well (Tab. 10 in Supp).



refs:

[1]  Whitened CLIP as a Likelihood Surrogate of Images and Captions, Betser et al. ICML 25.

[2] The Double-Ellipsoid Geometry of CLIP. Levi et al. ICML 25.

[3] On the Importance of Embedding Norms in Self-Supervised Learning. Draganov et al. ICML 25.

[4] CLIP-like Model as a Foundational Density Ratio Estimator. Uchiyama et al. Arxiv (very new indeed, reasonable that you weren't familiar with)

[5] General and Domain-Specific Zero-shot Detection of Generated Images via Conditional Likelihood. Betser et al. WACV 26'.

[6] Mind the gap: Understanding the modality gap in multi-modal contrastive representation learning. Liang et al. Neurips 22.

---

> ### Author Rebuttal · Authors · 2026-03-30
>
> We thank the reviewer for the thorough feedback and recognition that the direction is promising.
>
> **W1: Missing references**
>
> We agree they are highly relevant and will add them to the related work. While these papers study the representation space from different perspectives and do partially capture model confidence, REPVLM's distinct contribution is formalizing embedding density as an epistemic uncertainty proxy, and employing Riemannian FM to model it. Although relevant, Modality Gap [5] does not directly affect our approach, since REPVLM estimates density per-modality learning modal-conditioned probabilities.
>
> **W2: Additional baselines**
>
> We added Embedding Norm [3], W-CLIP [1], Guassian Mixture Model, and Conformity [2] on with results reported [**here**](https://anonymous.4open.science/r/icml26_rebuttal-00EF/clip.png). Density ratio estimator[4] is excluded because it measures how much an input constrains the cross-modal conditional, which captures input ambiguity (aleatoric) rather than model ignorance (epistemic).
>
> REPVLM consistently achieves the best or near-best performance. Notably, all competitive methods perform density estimation with varying tools explicitly or implicitly: REPVLM uses Riemannian FM, W-CLIP explicitly uses whitening + Gaussian approximation, Conformity implicitly uses cosine similarity between the input embedding and the embedding mean of a certainty modal, equivalent to von Mises Fisher Distribution approximatin, and GMM uses Gaussian mixtures. The ranking (REPVLM > W-CLIP > Conformity > GMM > Emb.Norm) aligns with density estimation quality and geometric awareness, supporting our theoretical framework.
>
> **W3: Limited VLM architectures**
>
> Our additional results on CLIP `vit-large-patch14`  with [**the results**](https://anonymous.4open.science/r/icml26_rebuttal-00EF/clip-L.png) confirmed the effectiveness of REPVLM. We acknowledge that REPVLM's theoretical guarantee requires $\ell_2$-normalized embeddings on $\mathbb{S}^{d-1}$ and the encoder geometry is shaped by semantic alignment: the $p(x) \to p(z)$ link in Section 3.1 relies on this. [**Results on SigLIP2**](https://anonymous.4open.science/r/icml26_rebuttal-00EF/siglip2.png) confirms this boundary: REPVLM maintains strong performance on distribution-shift benchmarks but degrades on ID benchmarks (ImageNet-1K, Food101, Cifar100). SigLIP2's multi-objective training (self-distillation, local feature extraction) injects texture-level information into the embedding, so density no longer purely reflects semantic coverage. For OOD inputs, unusual both semantically and texturally, density remains reliable for semantics-based classification. For ID inputs, where errors arise from class confusion, texture-driven density variation obscures the uncertainty–error correlation, analogous to how pixel-space generative models fail at OOD detection by capturing low-level statistics rather than semantics [6]. However, CLIP/SigLIP remain the most widely deployed VL encoders in practice, making REPVLM applicable to an impactful model class. We will scope our claims accordingly and discuss extending the framework to richer training objectives in future directions.
>
> **Minor W1:**
>
> Our statement refers to normalized embeddings, which lie on $\mathbb{S}^{d-1}$ by definition. We will clarify this.
>
> **Minor W2:**
>
> As the reviewer notes, this ceiling can be derived from the base accuracy at 0% rejection. We will add explict oracle accuracy curves in Figure 2 in the camera-ready. Regarding the higher variance on SigLIP, while the Spearman correlation shows more variance, REPVLM remains the only method with consistently positive and high correlation across all configurations.
>
> **Q1: Domain shift**
>
> Since VLMs in the paper are trained on web-scale data, their embedding geometry captures broad semantic structure. Domain-shifted inputs mapped to familiar semantic regions receive appropriate scores regardless of the proxy dataset, as long as the proxy dateset reflects the VLM's pretraining domain. This is confirmed by strong performance on ObjectNet, ImageNet-R, and ImageNet-Sketch.
>
> **Q2: Classification protocol**
>
> We use standard zero-shot classification: each class is encoded as "a photo of a [class name]", and classification selects the class with highest cosine similarity to the image embedding.
>
> **References**
>
> [1] Betser R., et al. Whitened CLIP as a Likelihood Surrogate of Images and Captions. *ICML*, 2025.
>
> [2] Levi, M, and Guy G., The Double-Ellipsoid Geometry of CLIP. *ICML*, 2025.
>
> [3] Draganov, A., et al. On the Importance of Embedding Norms in Self-Supervised Learning. *ICML*, 2025.
>
> [4] Uchiyama, F., et al. CLIP-like Model as a Foundational Density Ratio Estimator. *Arxiv preprint*, 2025.
>
> [5] Liang, V., et al. Mind the gap: Understanding the modality gap in multi-modal contrastive representation learning. *NeurIPS*,  2022.
>
> [6] Nalisnick, E., et al. Do Deep Generative Models Know What They Don't Know? *ICLR*, 2019.

---

> > ### Author Rebuttal · Reviewer_StS3 · 2026-04-01
> >
> > The paper, as I said before, is on timely and highly relevant topic. The additional refs are more than necessary to be added. I think that the comparison against current measurements is important, and show on which cases your method is superior, and where it is less (interesting why). Upon additions, and clarifications my concerns are resolved.

---

> > > ### Author Response · Authors · 2026-04-01
> > >
> > > We sincerely thank you for the thorough and constructive feedback throughout the review process. Your comments have been invaluable in strengthening the manuscript. We appreciate that the recommended references also enable a more complete interpretation of competing methods through the lens of density estimation and better position our contribution. The observed relatively less superior performance on SigLIP 2 is indeed intriguing and is a direction to be further studied in the consequent work.

---

### Official Review · Reviewer_SyRF · 2026-03-17

**Soundness:** 2
**Presentation:** 2
**Significance:** 2
**Originality:** 2
**Overall Recommendation:** 3
**Confidence:** 5

**Summary:**

The paper introduces a post-hoc technique for epistemic uncertainty estimation in frozen pretrained VLMs. The method aims to outline a way to estimate epistemic uncertainty by using density estimation on the embedding hypersphere. The use of Riemannian Flow Matching rather than Euclidean flow modeling. They also use the negative log-density $-log p(z)$ of embeddings as an uncertainty score, which is simple, interpretable, and model-agnostic. The formulation is appealing because it avoids expensive ensemble-style uncertainty estimation and instead leverages the geometry of the learned representation space.

**Compliance With Llm Reviewing Policy:**

Affirmed.

**Final Justification:**

I would like to thank the authors for their efforts in providing a rebuttal. I appreciate the answers to my questions. However, my overall conclusion remains unchanged. The main theoretical justification still seems heuristic, particularly the connection between input density and embedding density. Additionally, the sensitivity to proxy-dataset mismatch is not fully quantified. I believe the claims would be stronger if the theory were presented with more caution and if the practical assumptions were stated more explicitly. Therefore, I will maintain my current score.

**Key Questions For Authors:**

I have two fundamental questions and three questions regarding theoretical proof, and I would appreciate it if the authors could provide answers during the rebuttal.
* Q1: How should epistemic uncertainty be defined in a joint vision–text embedding space, rather than only through modality-conditioned marginals $p(z|c)$? The method models a shared hyperspherical embedding space but ultimately scores uncertainty as $-log p(z|c)$, with separate conditional target distributions for image and text embeddings. This raises a conceptual question: if the semantic space is genuinely joint, should epistemic uncertainty be tied only to modality-specific density, or should it also reflect cross-modal consistency between paired image/text representations? Put differently, can a sample be high-density under p(z|image) or p(z|text) while still being epistemically uncertain because the two modalities are misaligned in the shared space

* Q2: Can the authors decompose epistemic uncertainty into modality-specific and alignment-specific components? Since the model is trained on image-caption pairs but estimates uncertainty using modality-conditioned densities, it would be valuable to ask whether one can decompose uncertainty into terms such as U_ep = U_image + U_text + U_align, where U_image captures uncertainty from sparse visual regions, U_text captures uncertainty from sparse textual regions, and U_align captures uncertainty arising from weak or unstable image–text correspondence in the shared embedding space. Such a decomposition seems especially relevant because the current formulation appears to quantify only within-modality density, while many failure modes in VLMs are fundamentally cross-modal. How would such a decomposition be defined, estimated, and empirically validated?

* Q3: How should epistemic uncertainty be defined in a joint vision–text embedding space, rather than only through modality-conditioned marginals p(z|c)? The method models a shared hyperspherical embedding space but ultimately scores uncertainty as -log p(z|c), with separate conditional target distributions for image and text embeddings. This raises a conceptual question: if the semantic space is genuinely joint, should epistemic uncertainty be tied only to modality-specific density, or should it also reflect cross-modal consistency between paired image/text representations? Put differently, can a sample be high-density under p(z|image) or p(z|text) while still being epistemically uncertain because the two modalities are misaligned in the shared space

* Q4: The link from epistemic uncertainty to embedding density seems heuristic rather than formally derived: while the local linearization argument justifies a dependence of $U_{ep} (x)$ on the parameter Jacobian, it is unclear why this should imply that $−log p(z)$is a principled estimator rather than only an empirical proxy.

* Q2: How exactly does the first-order optimality condition imply a small Jacobian norm in high-density regions? The condition $E_{x ~ p(x)}[(∇_z ℓ)^T J_θ(x)] = 0$ is only an expectation-level statement, but the manuscript appears to infer pointwise suppression of $||J_θ(x)||_F$ in regions where $p(x)$ is high. This implication is not obvious and seems to require substantially stronger assumptions.

* Q5: The change-of-variables step appears mathematically delicate for VLM embeddings. Since the encoder maps high-dimensional inputs to normalized embeddings on the hypersphere, it is unclear in what precise sense the formula $p(z) = p(x) |det J_x(x)|^{-1}$ is valid here, given that the map is generally non-invertible and changes dimension.

**Strengths And Weaknesses:**

## Strengths
Overall, the method is simple and interpretable. The novelty and strengths from my side are:
* S1: Instead of modeling embeddings in Euclidean space, the method explicitly uses the fact that normalized VLM embeddings lie on the hypersphere $S^{d-1}$. The paper proposes estimating density directly on that manifold, which is more geometrically appropriate than standard Euclidean density modeling.
* S2: The proposed REPVLM framework learns a single conditional Riemannian Flow Matching model with a modality indicator $c$, so one model handles both image and text embedding distributions. That unified conditional formulation is a concrete methodological novelty of the paper.

## Weakness
* W1: The statement conflates epistemic uncertainty with embedding geometry and therefore overclaims the contribution: epistemic uncertainty is not an intrinsic geometric property of the embedding manifold, but a model-dependent quantity tied to uncertainty over parameters; at best, the paper argues that embedding density may serve as a proxy for epistemic uncertainty under additional assumptions, so calling it a “direct measure of confidence” is too strong!!

* W2: The method is described as task-agnostic, but it still requires a representative proxy dataset to estimate embedding density well. The paper acknowledges degradation under domain gap, meaning practical performance may depend strongly on the choice of proxy data.

* W3: Some reported empirical results seem unusually strong, for example, repeated near-perfect Spearman correlation values of 1.000 across many settings may raise concerns about metric sensitivity, threshold granularity, or whether the evaluation protocol is too favorable. More scrutiny and broader metrics would help validate the robustness of the claims.

* W4: Low density in embedding space may indicate rarity, long-tail structure, or demographic underrepresentation rather than genuine model ignorance. However, your paper briefly notes fairness risks, but this issue is more central than the discussion suggests and may limit the interpretation of density as epistemic uncertainty, this can be because of a mismatch between training data density and true epistemic uncertainty!

* W5: The process linking epistemic uncertainty to the parameter Jacobian norm, then to input density, and finally to embedding density relies on several strong assumptions. These include local linearity, the suppression of the Jacobian norm in high-density regions due to training, and the near-isometric behavior of the encoder (where the Jacobian norm is expected to decrease in high-density regions—this is not proven, only heuristically suggested). While these steps may seem intuitively plausible, the paper does not establish them with sufficient rigor to support the strength of its claims. Indeed, while the theoretical motivation may appear reasonable as an intuitive argument, it falls short as a rigorous derivation.

---

> ### Author Rebuttal · Authors · 2026-03-30
>
> We thank reviewer SyRF for their constructive feedback.
>
> **W1: Overclaiming the Contribution**
>
> We agree that embedding density is an *empirical proxy* for epistemic uncertainty, not a direct measure in the formal Bayesian sense. We will replace "direct measure of confidence" with "principled proxy".
>
> **W2: Proxy Dataset Dependence**
>
> Our method is task-agnostic (no labels needed) but requires a proxy dataset reflecting the VLM's pretraining domain. For general-purpose VLMs, our proxy datasets (CC, DataComp, LAION) naturally satisfy this; for specialized VLMs, a domain-appropriate proxy is needed, analogous to a calibration set in UQ. All results are consistent across all three proxy datasets. We will expand the Limitations section accordingly.
>
> **W3: Unusually Strong Results**
>
> The $S = 1.000$ values arise naturally: Spearman correlation is computed between rejection fraction and accuracy, and when uncertainty estimates are well-calibrated, accuracy increases monotonically with rejection, yielding perfect rank correlation. Importantly, not all methods saturate this metric: ProbVLM and MCDO frequently fall well below 1.0, demonstrating the metric does discriminate. As suggested by other reviewers, we have added additional baselines with the results reported [**here**](https://anonymous.4open.science/r/icml26_rebuttal-00EF/clip.png), none of which achieve consistently near-perfect $S$, further validating our method. It is also a standard metric used in UQ in previous works [1, 2, 3]. We also report Acc@90% Rejection as a complementary metric that differentiates methods even when $S$ values coincide.
>
> **W4: Fairness and Underrepresentation**
>
> We agree this deserves prominent treatment. A concrete risk is that practitioners using REPVLM for data curation could inadvertently filter out underrepresented groups. We will expand the Limitations section: when demographic metadata is available, uncertainty scores should be evaluated per subgroup. Without such metadata, high-uncertainty flags should be treated as candidates for human review rather than automatic removal.
>
> **Q1–Q3: Joint Embedding Space and Cross-Modal Alignment**
>
> Our method captures per-modality epistemic uncertainty: the model's confidence in its representation of a given image or text. Cross-modal alignment is a distinct measure: epistemic uncertainty asks "How much does the model know this input?", while alignment asks "do these two inputs correspond?" The former is a property of a single embedding, while the latter is a relational property of a pair.
>
> For an image-text pair, while our framework already provides $U_{\text{image}}$ and $U_{\text{text}}$ separately, an alignment component $U_{\text{align}}$ is conceptually appealing but nontrivial. Alignment depends on the joint configuration of both embeddings, and an additive decomposition may not be appropriate. In our opinion, there are two distinct approaches: one could extend REPVLM to estimating $p(z_{\text{img}} \mid z_{\text{txt}})$ or $p(z_{\text{img}}, z_{\text{txt}})$; alternatively, following BayesVLM's propagation of aleatoric uncertainty [2] through cosine similarity, one could similarly propagate our epistemic estimates to obtain a confidence score for the matching. We will discuss this as future directions.
>
> **W5, Q4–Q5: Theoretical Chain Assumptions**
>
> We acknowledge the theoretical chain is best understood as a motivating argument rather than a formal proof, and will revise Section 3.1 to explicitly label each step as formal derivation vs. empirical assumption.
>
> 1. **$U_\text{ep}$ → Jacobian Norm**: Standard assumption in Bayesian deep learning (e.g., Eq. 10 in [5]).
>
> 2. **Jacobian Norm → $p(x)$**: We fully agree with the reviewer that the condition is expectation-level and emphasize this in the manuscript. Additionally we confirmed this by computing CLIP's Jacobian norm via Hutchinson's random projection for in-distribution (ImageNet-1K) and OOD (EuroSAT) samples: ImageNet-1K mean $24.49$ (median $13.54$) vs. EuroSAT mean $47.99$ (median $32.19$). Despite long tails, a clear statistical tendency exists.
>
> 3. **$p(x)$ → $p(z)$**: VLM encoders are not strict isometries, but contrastive objectives encourage alignment and uniformity, promoting near-volume-preserving local behavior. You could refer to our detailed justification to Reviewer 4 (S4Jo) addressing SW4 & Q5.
>
> **References**
>
> [1] Upadhyay, U., et al. Probvlm: Probabilistic adapter for frozen vison-language models. *ICCV*, 2023.
>
> [2] Venkataramanan, A., et al. Probabilistic Embeddings for Frozen Vision-Language Models: Uncertainty Quantification with Gaussian Process Latent Variable Models. *UAI*, 2025.
>
> [3] Gomez, L. Over the Top-1: Uncertainty-Aware Cross-Modal Retrieval with CLIP. *UAI*, 2025.
>
> [4] Baumann, A., et al. "Post-hoc probabilistic vision-language models." *ICLR*, 2026.
>
> [5] Immer A., et al. Improving predictions of Bayesian neural nets via local linearization. *AISTATS*, 2021.

---

> > ### Author Rebuttal · Reviewer_SyRF · 2026-04-03
> >
> > Thank you for your rebuttal. I appreciate your answers to my questions. The revisions help, particularly the clearer framing of embedding density as a proxy for epistemic uncertainty and the clarification that the method captures per-modality uncertainty rather than cross-modal alignment. However, my overall conclusion remains the same because **the main theoretical justification still appears heuristic**, especially the link from input density to embedding density, and the sensitivity to proxy-dataset mismatch is still not fully quantified. I believe the work is interesting and promising, but the claims would be stronger if the theory were presented more cautiously and the practical assumptions were stated more explicitly. Therefore, i would keep my score as it is!

---

> > > ### Author Response · Authors · 2026-04-04
> > >
> > > We thank the reviewer for their continued engagement. We would like to highlight two points regarding the strength of the existing evidence:
> > >
> > > **On the theoretical chain $p(x) \to p(z)$**
> > >
> > > We agree that Section 3.1 should more cautiously separate formal steps from empirically motivated ones and will revise accordingly. However, we wish to emphasize that establishing a rigorous density-preservation result for contrastive encoders is a fundamental open problem in the broader representation learning community — not an oversight specific to our work. Recent theoretical efforts illustrate both the difficulty and active interest: [1] show that under certain alignment and concentration assumptions, projections of the high-dimensional representation asymptotically approach a multivariate Gaussian distribution; [2] prove InfoNCE-trained encoders implicitly invert the data generating process under generative assumptions; [3] develop a measure-theoretic framework characterizing how InfoNCE shapes representation measures on the embedding manifold. Moreover, [4] provides independent empirical confirmation that CLIP embedding density serves as a meaningful likelihood surrogate - a finding that only holds because the density correspondence approximately holds in practice. A complete formal proof would itself constitute a major standalone contribution, resolving a foundational problem that the theory community has not yet settled and effectively grounding aspects of the Platonic Representation Hypothesis [5].
> > >
> > > Our contribution is to identify this connection, ground it in established and emerging theoretical evidence, build a practical framework around it, and validate it empirically. Notably, the SigLIP2 analysis (response to Reviewer StS3) demonstrates that our framework correctly predicts its own failure modes when the contrastive assumptions are violated — providing evidence that the identified theoretical link is not merely correlational.
> > >
> > > **On proxy-dataset sensitivity**
> > >
> > > The need for representative reference data is structural to virtually all post-hoc UQ methods — temperature scaling, isotonic regression, and conformal prediction all require held-out calibration data and degrade under domain mismatch [6]. This is not unique to REPVLM. Our experiments also address the robustness of REPVLM: consistent performance across three proxy datasets of different composition, and strong results on distribution-shift benchmarks where baselines frequently fail.
> > >
> > > **References**
> > >
> > > [1] Betser, Roy, et al. "InfoNCE Induces Gaussian Distribution." ICLR, 2026.
> > >
> > > [2] Zimmermann, Roland S., et al. "Contrastive learning inverts the data generating process." ICML, 2021.
> > >
> > > [3] Cai, Yichao, et al. "The Geometric Mechanics of Contrastive Representation Learning." arXiv preprint arXiv:2601.19597, 2026.
> > >
> > > [4] Betser, Roy, et al. "Whitened CLIP as a Likelihood Surrogate of Images and Captions." ICML, 2025.
> > >
> > > [5] Huh, Minyoung, et al. "The platonic representation hypothesis." ICML, 2024.
> > >
> > > [6] Tomani, Christian, et al. "Post-hoc uncertainty calibration for domain drift scenarios." CVPR, 2021.

---

### Decision · Program_Chairs · 2026-04-30

**Decision:**

Accept (regular)

**Comment:**

This paper introduces a post-hoc technique for epistemic uncertainty quantification in vision-language models (VLMs). The method leverages Riemannian Flow Matching to perform density estimation directly on the manifold of embeddings. By using the negative log-density of these embeddings as an uncertainty score, the framework provides a model-agnostic and alternative to traditional ensemble-based uncertainty estimation.

Reviewers largely praised the paper's choice of topic as interesting and timely, with a technically good method that innovates. The reviewers had several questions and issues in the initial reviews including:
- Reliance on strong, unproven assumptions (local linearity, isometry, etc) to link Bayesian epistemic uncertainty to embedding density
- Lack of theoretical rigor when linking the above concepts
- Missing cites and comparisons to common approaches like GMM, Embedding Norm, etc
- Unusually strong results for correlations
- Limited number of VLMs tested
- Sensitivity to choice of domain, and robustness to OOD inputs or concepts that could occur in real-world conditions.

Many of these issues were satisfactorily resolved during the discussions.
- While the paper does indeed rely on strong assumptions for the theoretical connections to hold, they have been discussed. The authors present some tests of the extent to which the assumptions hold in practice, but there is room for expansion.
- There is some lack of rigor, but the author will clarify in the text when they are making intuitive connections vs. claiming exact correspondence
- The authors added comparisons to four baselines methods and will discuss them in the paper. The conclusions remain unchanged
- Some strong correlations were explained, and do not seem to indicate any problem with the evaluations
- The breadth of VLMs tested remains outstanding. The authors should be cautious to not over-claim how broadly their results apply across today's VLMs when only two architectures have been tested.
- Some testing on OOD data was already in the paper, but there is room for expanded discussion on this point. The reviewers recommend that the authors discuss the assumption that there is a reasonably representative proxy distribution, which might not hold in practice, especially for more specialized or rapidly changing domains.

Overall I believe the reviewers have done a great job at analysing this work, asking important questions, and the authors have effectively argued that their paper's results stand. Hence I am recommending acceptance.

The authors must update their paper with the promised changes, experiments, and caveats to make clear the limitations of their work, and especially the theoretical connections.